# The auxin phenylacetic acid induces *NIN* expression in the actinorhizal plant *Datisca glomerata*, whereas cytokinin acts antagonistically

Marco Guedes Salgado [1¤a] *, Pooja Jha Maity[1¤b], Daniel Lundin [2,3], Katharina Pawlowski [1] *

1 Department of Ecology, Environment and Plant Sciences, Stockholm University, Stockholm, Sweden,
2 Centre for Ecology and Evolution in Microbial Model Systems, Linnaeus University, Kalmar, Sweden,
3 Department of Biochemistry and Biophysics, Stockholm University, Stockholm, Sweden

¤a Current address: Institute of Biotechnology, Helsinki Institute of Life Sciences, University of Helsinki, Helsinki, Finland
¤b Current address: Department of Botany, Hansraj College, University of Delhi, Delhi, India
* katharina.pawlowski@su.se (KP); marco.salgado@helsinki.fi (MGS)

**Data Availability Statement:** All the cDNA sequences used in this study were deposited in GenBank under the accessions listed in S1 Table.

## Abstract

All nitrogen-fixing root nodule symbioses of angiosperms–legume and actinorhizal symbioses–possess a common ancestor. Molecular processes for the induction of root nodules are modulated by phytohormones, as is the case of the first nodulation-related transcription factor *NODULE INCEPTION* (*NIN*), whose expression can be induced by exogenous cytokinin in legumes. The process of actinorhizal nodule organogenesis is less well understood. To study the changes exerted by phytohormones on the expression of the orthologs of *CYCLOPS*, *NIN*, and *NF-YA1* in the actinorhizal host *Datisca glomerata*, an axenic hydroponic system was established and used to examine the transcriptional responses (RT-qPCR) in roots treated with the synthetic cytokinin 6-Benzylaminopurine (BAP), the natural auxin Phenylacetic acid (PAA), and the synthetic auxin 1-Naphthaleneacetic acid (NAA). The model legume *Lotus japonicus* was used as positive control. Molecular readouts for auxins and cytokinin were established: *DgSAUR1* for PAA, *DgGH3.1.* for NAA, and *DgARR9* for BAP. *L. japonicus NIN* was induced by BAP, PAA, and NAA in a dosage- and time-dependent manner. While expression of *D. glomerata NIN2* could not be induced in roots, *D. glomerata NIN1* was induced by PAA; this induction was abolished in the presence of exogenous BAP. Furthermore, the induction of *DgNIN1* expression by PAA required ethylene and gibberellic acid. This study suggests that while cytokinin signaling is central for cortex-induced nodules of *L. japonicus*, it acts antagonistically to the induction of nodule primordia of *D. glomerata* by PAA in the root pericycle.

## Introduction

Root nodule symbioses between plants and nitrogen-fixing soil bacteria are restricted to species from four orders of dicots: Fabales, Fagales, Cucurbitales and Rosales. Unicellular Gram-

**Funding:** This work was supported by a grant from the Swedish Research Council Vetenskapsrådet (VR 2012-03061 to KP) and a grant from Carl Tryggers Stiftelse för Vetenskaplig Forskning (CTS 13:354 to KP). The funders had no role in study design, data collection and analysis, decision to publish, or preparation of the manuscript.

**Competing interests:** The authors have declared that no competing interests exist.

**Abbreviations:** AVG, L-alpha-(2-aminoethoxy vinyl) glycine; GA, gibberellic acid; IAA, indole-3-acetic acid; NAA, 1-naphtaleneacetic acid; PAA, phenylacetic acid; PBZ, paclobutrazol.

negative soil bacteria, collectively called rhizobia, induce the formation of nodules on roots of legumes (Fabales) and, exceptionally, on those of a non-legume, *Parasponia* sp. (Cannabaceae, Rosales). Filamentous Gram-positive soil actinobacteria from the genus *Frankia* induce nodules on roots of 25 genera from eight families of the Fagales, Rosales, and Cucurbitales, collectively called actinorhizal plants [1]. Both symbioses possess a common ancestor and are restricted to a single clade, the Fabids [2]. Phylogenetic studies indicate that the ancestor of Fabids was symbiotic [2–4], however proof of whether this ancestor could already form root nodules remains to be shown [5]. While legume symbioses are essential in agriculture insofar as they yield protein-rich seeds while rendering their host plants with an independent source of nitrogen fertilizer, thereby contributing to increases of nitrogen pools in surrounding ecosystems, actinorhizal species mostly represent pioneer plants and are often used in reforestation or soil recovery [6].

In all these interactions, the microsymbionts are not vertically transmitted; although the microsymbionts can exceptionally be carried by host-produced seeds [7], nodule induction always starts with bacteria in the soil, not with bacteria in the plant. Thus, each plant must be colonized *ab initio* in a process that requires signal exchange. Rhizobia produce lipochitooligosaccharide (LCO) nodulation factors (Nod factors; [8]) to induce the response of early nodulins genes. In comparison with model legumes, whose response to Nod factors have been analysed in detail, and enabled the characterization of many components of the Nod factor signal transduction pathway [9], in actinorhizal plants the key molecular components remain less understood. In fact, in actinorhizal plants, only a few genes encoding enzymes involved in nodule induction have been identified and characterized and the foundation of these studies was based on direct homology with the corresponding legume proteins [10]. On the microsymbiont side, there is indication that most *Frankia* strains do not produce LCO Nod factors [11]. Notwithstanding, homologs of the rhizobial canonical *nod* genes *nodABC*, which code for enzymes committed with steps for the assembly of LCO backbone, are present in two strains of the earliest branching *Frankia* clade and are expressed in symbiosis [12, 13]. *Frankia* strains infecting *Datisca glomerata* and *Datisca cannabina* (Datiscaceae, Cucurbitales) contain *nodABC* genes, however experimentation with these strains remains hampered by the impossibility of culturing them, making it challenging to address questions concerning the symbiotic signalling associated with these ancient symbioses.

The involvement of phytohormones in legume nodule induction has been examined in detail. Cytokinin signaling is central to nodule organogenesis and it has been shown that the external application of cytokinin or an autoactivated form of a cytokinin receptor can be sufficient to induce the formation of legume nodules in the absence of rhizobia; however, cytokinin has a negative effect on infection [14–18]. It has to be pointed out that in the only non-legumes nodulated by rhizobia, *Parasponia* spp., the involvement of cytokinin signalling in nodule organogenesis is not yet clear since knockout mutants of one cytokinin receptor are not affected in nodulation; this may be a result of receptor redundancy–or not [19]. The role played by cytokinin in nodulation has also changed when the intercellular infection pathway of the model legume *Lotus japonicus* was examined, in that knockout mutants of the cytokinin receptor showed a much stronger phenotype than those used for intracellular infection studies [20]. Multiple studies have shown the involvement of auxin signaling not only in the induction of legume nodules but also on their actinorhizal counterparts [21–24]. Altogether, it is widely accepted that the interplay of cytokinin and auxin signalling is required to form a local and polarized auxin maximum, essential to trigger cell division and nodule primordium formation [25–27]. Gibberellin signalling mediated by DELLA proteins interacting with different transcription factors underlines a positive and a negative role: gibberellins promote nodule organogenesis [28], but inhibit nodule infection [29–32]. Controversial roles were also discussed for

ethylene: while negative effects were reported on induction of legume nodule primordium and infection thread formation (reviewed by [33]), positive effects were shown for ethylene governing the transport of auxin from the shoot to the nodulation zone [34]. Furthermore, a positive effect for ethylene was also observed on the nodulation of semi-aquatic legumes under flooded conditions [35].

Phytohormone signalling is involved in several steps of legume nodule induction [36]. Expression of the first transcription factor specifically involved in nodulation, *Nodule Inception* (*NIN*), is induced by Nod factor signalling via $Ca^{2+}$ spiking sensed by the calmodulin-dependent protein kinase CCaMK which interacts with the transcription factor CYCLOPS [37]; *NIN* expression can also be induced by exogenous supply of cytokinin in *L. japonicus* [17, 18]; NIN acts as a central regulator and is crucial to induce the expression of downstream nodule-specific transcription factors, such as the CCAAT box-binding NF-Y transcription factor NF-YA1 [38–40]. Previous studies indicate that *NIN* is involved in nodule organogenesis and infection in the actinorhizal species *Casuarina glauca* (Casuarinaceae, Fagales) [11, 41]. *D. glomerata* is one of the few symbiotic species known so far whose genome contains two copies of *NIN* [3]. Promoter sequence analysis of *D. glomerata NIN1* led to the identification of multiple cytokinin- and auxin-responsive elements [42–45], suggesting that *DgNIN1* signaling might depend on auxin and/or cytokinin.

Nodules of *D. glomerata* differ from those of *L. japonicus* in anatomy, physiology, and ontology [46]. To understand whether the symbiotic programs of these two species share conserved mechanisms for nodule induction, which might unravel conserved traits across far related lineages, we set out to investigate the effects exerted by auxin and cytokinin on expression of *D. glomerata NIN1*, while substantial information already existed for *L. japonicus NIN* [17, 47, 48]. The analysis was extended to the orthologous genes *DgCYCLOPS* and *DgNF-YA1*, which, based on direct inference from research carried on model legumes, act upstream and downstream of *DgNIN1*, respectively; *DgNF-YA1* codes for the CCAAT box-binding transcription factor NF-YA1 and its promoter is transactivated by NIN in model legumes [38–40].

Using an axenic hydroponic system that facilitates the transfer of plantlets from one medium to another, gene expression studies (RT-qPCR) on *L. japonicus* and *D. glomerata* were performed on three genes which are induced consecutively during nodule induction in legumes: *CYCLOPS*, *NIN1*, and *NF-YA1*. Their expression in response to treatment with cytokinin and auxins was examined alone and in combination with inhibitors of gibberellin and ethylene signalling. The synthetic cytokinin 6-Benzylaminopurine (BAP) was chosen for this study because it had been used in several legume nodule induction studies [17, 49]. The decision for the type of auxin was onerous because the most common endogenous plant auxin, Indoleacetic Acid (IAA), is light sensitive and thus may lead to false negative results when applied in glass vials. The most commonly used synthetic auxin, 1-Naphthaleneacetic Acid (NAA), in contrast to natural auxins, diffuses through membranes in the absence of specific transporters, which, again, might distort the results [50]. Since the dominant auxin in roots of *D. glomerata* is not Indoleacetic Acid, but Phenylacetic Acid (PAA) [51], and significant amounts of PAA were also found in other actinorhizal plants [22, 23], special emphasis was given to PAA in this study. Since gibberellin and ethylene are involved in nodule induction, in particular ethylene whose effects are known to often overlap with those of cytokinin [18, 19, 33, 52, 53], the effects of the ethylene biosynthesis inhibitor aminoethoxyvinyl glycine (AVG) and those of the gibberellin biosynthesis inhibitor paclobutrazol (PBZ) were examined in this study.

With this work, we strove to extend to actinorhizal nodules the abundant body of knowledge that transpired from studies carried on model legumes to what concerns the role played by auxin, cytokinin, ethylene, and gibberellin signalling in legumes root nodules development.

## Results

### Establishment of a hydroponic system and identification of reporter genes for phytohormone responses in roots of *D. glomerata*

To analyse the effects of phytohormones on the expression of genes involved in nodule development, we established an axenic hydroponic system that served to exogenously challenge roots of *Lotus japonicus* and *Datisca glomerata* (Fig 1).

 *D. glomerata* contains two copies of *NIN* [3] and although both *NIN1* and *NIN2* are induced in nodules compared to roots, only *NIN2* is expressed nodule-specifically with no detectable levels of expression on roots of either seedlings or greenhouse-grown plants (Fig 2A; for *NIN1* see also [55]). Cathebras et al. [54] have shown the presence of the *cis*-regulatory element *PACE* within the promoter of *DgNIN2* (termed *DgNIN1* in [54]) as a requirement for induction by the CCaMK/CYCLOPS complex, whilst the promoter of *DgNIN1* contains an uncommon and extended version of *PACE*, which compared to that of *DgNIN2*, or those of other root nodule-forming plants, contains an insertion of eight nucleotides at the core of *PACE* and displays high nucleotide conservation across *PACE-Y* and *PACE-X*, and a fair conservation (*ca*. 67%) at the core of the *PACE* element (Fig 2B). *DgNIN1* combined with its PACE element could complement the nin15 mutant of *L. japonicus*, *i.e.*, this non-standard PACE element conferred induction by the CCaMK/CYCLOPS complex of *L. japonicus* [54]. To examine whether *DgNIN1 PACE* holds physical properties for distortion and to adopt a structural conformation to bend around a protein, inference about the DNA rigidity of the helices formed by

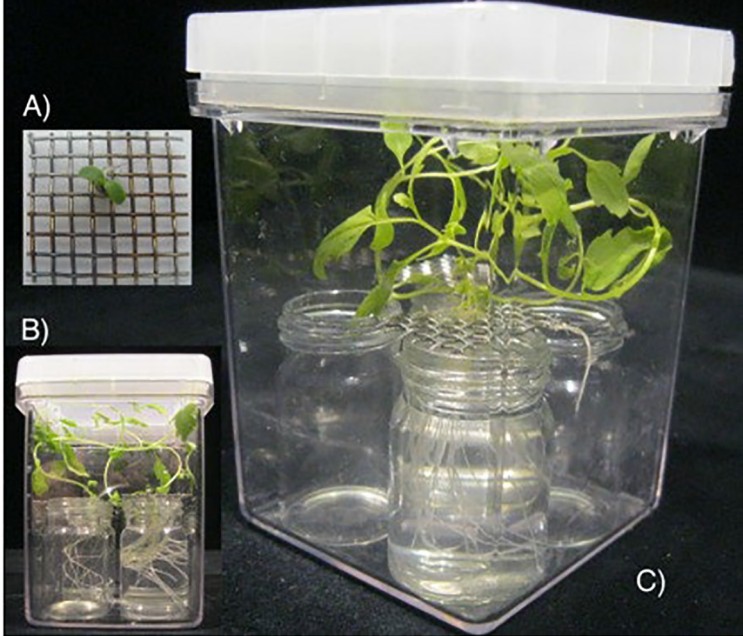

**Fig 1. Establishment of an axenic hydroponic system.** Seeds were surface-sterilized and imbibed with sterile MilliQ water at 4ºC for 5 days in the dark. After thorough washing, seeds were placed on 20x20 (mm) metal grids and allowed to germinate in a thin layer (*ca*. 2 mm) of ¼ strength Hoagland's medium [87] supplied with 10 mM nitrogen (Hoaglands N$^+$) and 0.8% plant agar (Duchefa). Panel A shows a detail of a recently germinated *Lotus japonicus* root. One week after germination, metal grids containing germinated seeds (like the one depicted in panel A) were placed over vials filled with 18 ml of Hoagland's N$^+$. This way, recently emerged root tips were able to develop within a fully controlled environment. Growing proceeded in Magenta boxes under white light [200 μEm$^{-1\ s^{-1}}$ m$^{-2}$ at 16h light/23˚C and 8h dark/18˚C]. Panels B and C show hydroponically treated roots of *Datisca glomerata*.

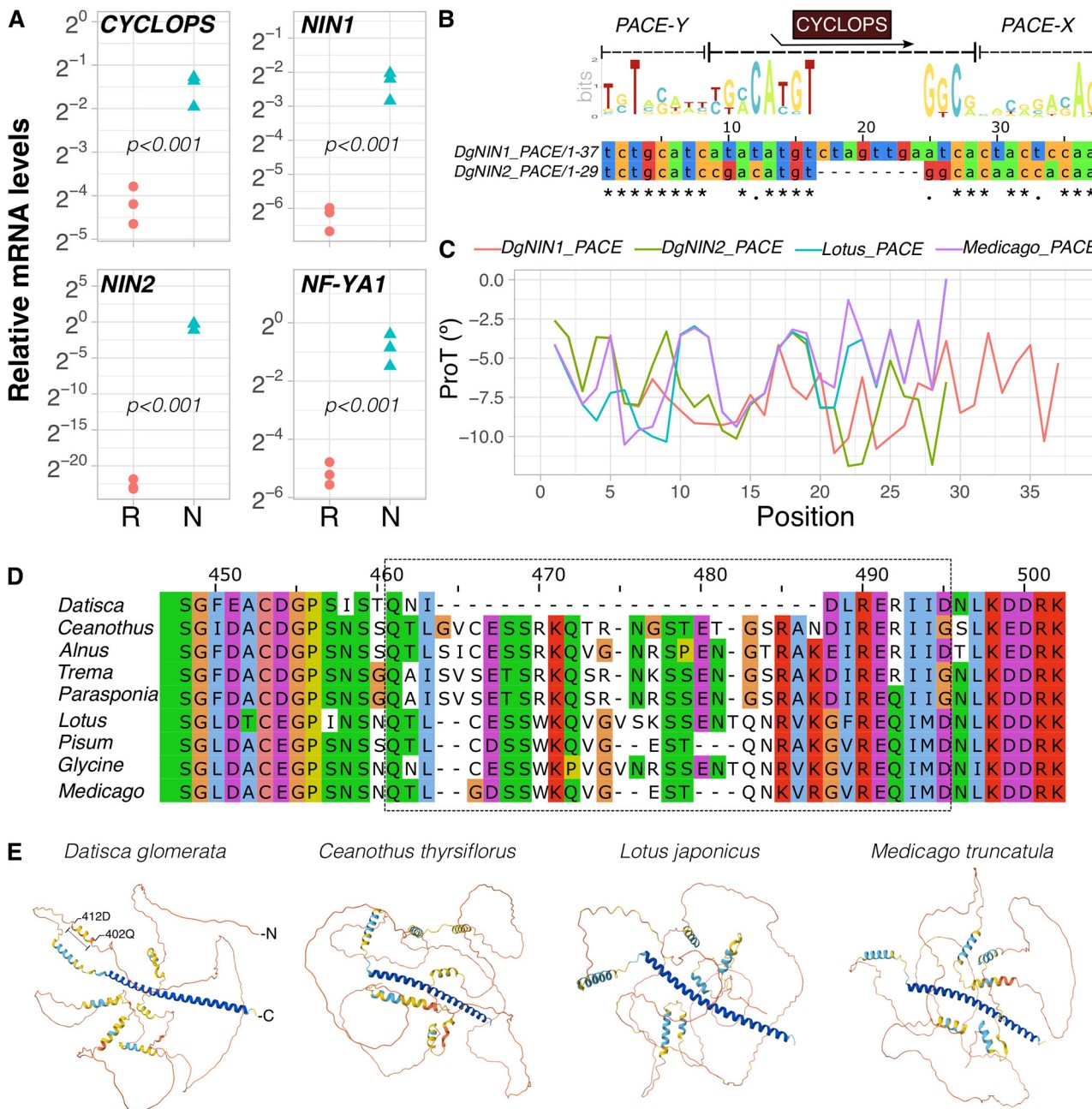

**Fig 2. The *PACE* elements of *DgNIN1* and *DgNIN2* put in context.** (A) Expression profile of genes encoding orthologs of nuclear transcription factors associated with nodule development in *Datisca glomerata*; transcript abundance was analysed by RT-qPCR in roots (R) and nodules (N) of greenhouse-grown plants and is given relative to that of the housekeeping gene *EF1*-α (n = 3 for both technical and biological replicates); differences between R and N are shown at p<0.001 (student's *t* test). (B) *In silico* analysis of the *cis*-regulatory element *PACE* in *DgNIN1* and *DgNIN2* promoters; representative logo profile of *PACE* depicted as position weight matrix out of 14 FaFaCuRo species and 2 non-nodulators (*Prunus persica* and *Ziziphus jujuba*) along with a comparison of *NIN PACE* sequences in *D. glomerata* showing an 8-nucleotide insertion at the core of *DgNIN1 PACE*. (C) Evaluation of secondary order effects by propeller-twist (ProT) of base-pairs across the *PACE* landscape of *D. glomerata* and model legumes. (D) Implications of primary sequence dissimilarity in secondary structure of CYCLOPS shown as 3D models. Partial alignment of CYCLOPS from 8 nodulating species and 1 non-nodulator (*Trema tomentosa*) shows dissimilarities in a region encompassing 24 residues (see dashed box) (full alignment in S2A Fig). (E) AlphaFold 3D models showing the implications in secondary structure of DgCYCLOPS at the site of the region covered by the dashed box, highlighting the presence of an α-helix in DgCYCLOPS, from 402Q to 412D, which is absent in *Ceanothus thyrsiflorus* and model legumes.

*PACE* was conducted by propeller-twisting (ProT) of base-pairs. The angle predicted for *DgNIN1 PACE* by ProT analysis was always lower than 11,06° across the landscape of 37 nucleotides, a measure that correlates with lower rigidity of the helices spanning the *PACE* sequences *DgNIN2* and those of the model legumes *L. japonicus* and *Medicago truncatula* (Fig 2C). To gain a better understanding of the *PACE*-CYCLOPS interactions in *D. glomerata*, we compared the amino acid sequence of DgCYCLOPS with those of CYCLOPS proteins from a range of root nodule-forming plants and one non-nodulator, *Trema tomentosa*. Alignment of CYCLOPS orthologs showed that the primary structure of CYCLOPS, along with its ortholog in *Ceanothus thyrsiflorus*, displays a ~40 amino acid sequence extension at its N-terminus that is neither present in *Alnus glutinosa*, *Parasponia andersonni*, nor in the legumes included in the analysis (S2A Fig); importantly, the first 20 amino acids represent a disordered domain whose presence at the N-terminus seems to be a feature of actinorhizal plants as it was only found in *Datisca glomerata* (position 1–20), *Alnus glutinosa* (position: 1–33) and *Ceanothus thyrsiflorus* (position 16–36) (S2A Fig). Furthermore, 17 amino acids upstream of the largest and well-conserved disordered domain, placed at the C-terminus, the primary structure of DgCYCLOPS shows high dissimilarity with that of other orthologs (Fig 2D; full alignment in S2A Fig). To investigate whether this dissimilarity could lead to alterations in protein secondary structure, AlphaFold structure predictions were carried out and the comparison of the predicted models showed the presence of an *α*-helix in *Datisca glomerata* that is absent in *Ceanothus thyrsiflorus*, *L. japonicus*, and in *Medicago truncatula* (Fig 2E).

Like for *M. truncatula NIN*, promoter sequence analysis of *D. glomerata NIN1* [3] showed the presence of multiple cytokinin- and auxin-responsive elements [42–45] (see https://doi.org/10.17045/sthlmuni.9275285.v1). To link transcriptional responses in roots to exogenous application of phytohormones, the genetic dependencies of the effects exerted by phytohormones had to be confirmed independently. Although marker genes for phytohormone responses are well established in several plant species, so far none were characterized in *D. glomerata*. To assess potential candidates that could serve as molecular readouts for phytohormone responses in *D. glomerata*, we conducted homology analysis by mining a previously published transcriptome [55, 56]. For cytokinin, three potential marker genes were identified from the orthologous group ENOG411BR8V and named after the two-component response regulator orthologs ARR4, ARR5, and ARR9 of *Arabidopsis thaliana*. Preliminary assays confirmed *ARR9* as a suitable marker gene for cytokinin in *D. glomerata*, owing to an increase of *ARR9* mRNA levels in the presence of BAP (see below).

To identify reporter genes for auxin, we searched for candidate members of the *Gretchen Hagen3* (GH3), Small Auxin Upregulated RNA (SAUR), and GATA families. Although some representatives of the GATA family of transcription factors could be identified in the nodule transcriptome of *D. glomerata* (GATA15-, 16-, 24-, and 26-like), none of these candidates were orthologous to auxin-responsive GATA23 of *Arabidopsis* [57]. However, a member of the Small Auxin Upregulated RNA (SAUR) gene family was identified based on the orthologous group ENOG410JVA8 and the encoded protein shares high sequence similarity with other relatives in Viridiplantae (SAUR1; S1A Fig). Furthermore, the hidden Markov model returned by the profile of AtGH3.5 enabled the identification of two proteins in *D. glomerata*, which we named GH3.1 and GH3.2. These proteins were analysed *in silico* for the presence of motifs previously linked with auxin responsiveness [58] and this analysis indicated that GH3.1 was very likely orthologous to the auxin responsive AtGH3.5/WES1, while GH3.2 was probably responsive to jasmonic acid [59] (S1B Fig). As a result, *SAUR1* and *GH3.1* were selected for downstream analysis as putative auxin markers in *D. glomerata*.

To validate the responsiveness of *GH3.1* and *SAUR1* to auxins, their transcript abundance was analysed by real-time quantitative PCR (RT-qPCR) on cDNA from roots treated with

NAA and PAA across a wide range of concentrations. Under the conditions tested in this experiment, only *GH3.1* showed a convincing transcriptional response to NAA (p = 0.09, 10 nM; p = 0.0021, 100 nM; p = 0.0014, NAA 500 nM; S2 Table; S1C Fig). Given that no differences were observed between 100 nM and 500 nM, this observation suggested that 100 nM may be already above the optimum for *GH3.1* response to NAA. Expression levels of *SAUR1* were neither affected by NAA nor by PAA, but the range of concentration tested for PAA in this assay (5nM—25nM) was very likely below the optimum (S1C Fig). Because PAA is the dominant auxin in roots of *D. glomerata* [51], we reasoned that the response of *SAUR1* should be examined in greater detail for a higher concentration of PAA (see below). Taken together, results showed *GH3.1* as a suitable candidate to address effects exerted by NAA, whereas *SAUR1* did not respond to NAA (10nM—500nM) nor to PAA (<25 nM) under the conditions applied in this assay.

## Use of the hydroponic system to address responses in gene regulation in roots exogenously treated with phytohormones: Proof of concept using the model legume *Lotus japonicus*

To assess the possibility of using the hydroponic system coupled with RT-qPCR to assess the effects of phytohormones on nodulation-related gene expression in *D. glomerata*, the effects exerted by phytohormones were first examined in the well-studied symbiotic model *L. japonicus*. The expression of *L. japonicus* cytokinin oxidase/dehydrogenase3 (*LjCkx3*; [60]) was induced by BAP when applied for a period of 8 h (p<0.01) and its effect persisted after 24 h at 100 nM (p<0.01; Fig 3). Auxin responsiveness was analysed based on the expression of the well-known *L. japonicus* auxin-responsive promoter *LjGH3* [61, 62], which in this assay was induced exclusively by NAA at 10 nM and 100 nM when applied for 8 h or 24 h (p<0.01). Yet, *LjGH3* suppression by 100 nM of BAP applied for 8 h (p<0.01) indicates that auxin and cytokinin act antagonistically on *LjGH3* expression in *L. japonicus* roots.

As for differential regulation of *LjCYCLOPS* and *LjNIN* expression in roots of *L. japonicus*, the expression level of *LjCYCLOPS* did not change under any of the conditions examined, in contrast with that of *LjNIN* which did change in response to cytokinin and auxins. The transcription of *LjNIN* was induced after 8 h of exposure to 100 nM of BAP (p<0.01) and this effect persisted after 24 h at both 10 nM and 100 nM (p<0.05). The auxin NAA induced changes of *LjNIN* expression in roots treated with 10 nM or 100 nM over a period of 8 h or 24 h, respectively (p<0.01). Likewise, *NIN* was induced by the natural auxin PAA, but only in roots exposed for 24 h to a concentration of 10 nM (p<0.05).

These results show that the cytokinin BAP is able to modulate the expression of *LjNIN* in a dosage- and time-dependent manner when exogenously applied to roots of 30-day-old seedlings grown hydroponically, which is in line with a previous study carried with roots grown in Petri dishes on agar containing BAP [15]. Thus, proof of concept was obtained.

## Nanomolar concentrations of the synthetic cytokinin BAP repressed the transcription of *CYCLOPS*, *NIN1*, and *NF-YA1* in 30-day-old roots of *D. glomerata* seedlings

The *D. glomerata* orthologs of *CYCLOPS* and *NF-YA1* were identified phylogenetically (S2 and S3 Figs). To address the question whether the phytohormone background required for nodule development in *D. glomerata* shares commonalities with that of *L. japonicus*, 30-day-old roots of *D. glomerata* were challenged with concentrations of BAP ranging from 10 nM to 500 nM, in 24 h assays. Unlike the transcriptional response observed in roots of *L. japonicus* (see Fig 3), increasing amounts of cytokinin led instead to repression in expression of both *CYCLOPS*

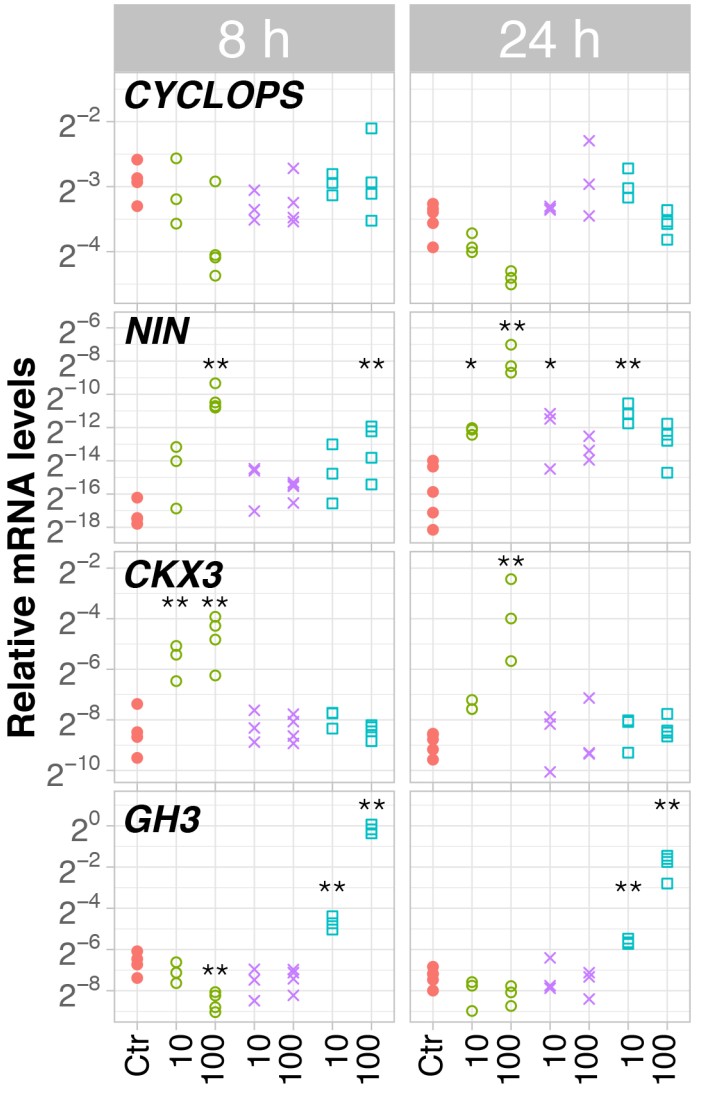

**Fig 3. *Lotus japonicus* transcriptional response to cytokinin (BAP) and auxins (PAA and NAA).** Transcript abundance was analysed by RT-qPCR after 30-day-old Gifu roots treated during 8h and 24h (technical replicates, n = 3). *Y*-axis shows the mRNA quantity relative to that of *PUQ*, *ATPs*, and *PP2A*. Phytohormones and molarities are given on the *X*-axis. With the exception of Ctr, 24h (n = 5), Ctr, 8h; PAA100, 8h; NAA100, 8h/24h (n = 4), the number of biological replicates was n = 3. Gene names are given. Significant differences to control are highlighted at p<0.05 (*) and p<0.01 (**).

(p = 0.09, 50 nM BAP; p<0.001 for both 100 nM BAP and 500 nM BAP) and *NIN1* (p = 0.032, 500 nM BAP) in a dosage-dependent manner (S2 Table; Fig 4). The transcriptional profile exhibited by *CYCLOPS* in this assay indicates that its transcription is tightly regulated by BAP levels within the range of 10 nM to 100 nM. Since the promoters of legume Nuclear Factor-Y (NF-Y) subunit genes are expected to be targeted and activated by NIN1 in analogy with legumes [38–40], we then set to investigate whether increases in BAP levels would be

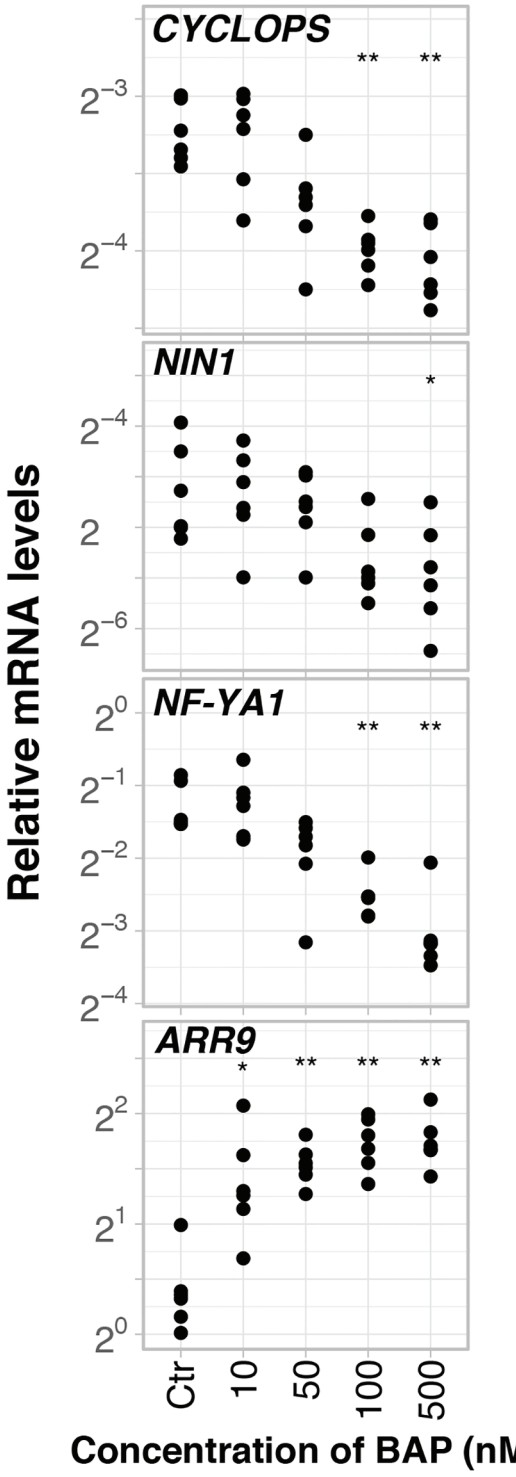

**Fig 4.** ***Datisca glomerata* transcriptional response to cytokinin (BAP).** Transcript abundance was analysed by RT-qPCR after 30-day-old roots treated during 24h (technical replicates, n = 3). *Y*-axis shows the mRNA quantity relative to that of *EF1-α*, *PUQ* and *TIP41*. BAP molarities are given on the *X*-axis (number of biological replicates, n = 6). Significant differences to the control are highlighted at p<0.05 (*) and p<0.01 (**). Gene names are given.

accompanied by changes in expression of *NF-YA1*. Notably, the transcriptional profile exhibited by *NF-YA1* was similar to that of *CYCLOPS* and *NIN1* insofar as the transcript abundance of *NF-YA1* decreased inversely with BAP levels (p<0.001, 100 nM and 500 nM; S2 Table; Fig 4).

## Transcription of *NIN1* and *NF-YA1* was induced by nanomolar concentrations of PAA in 52-day-old roots of *D. glomerata*

Since in *D. glomerata*, BAP acted antagonistically towards the expression of genes whose protein products act upstream of *NF-YA1* induction in model legumes, we hypothesized that either *D. glomerata* roots exhibit a different temporal window in which nodulation-related gene expression can be activated when compared with roots of *L. japonicus*, or that instead of cytokinin, another phytohormone was involved in regulating the expression of these nodulation-related transcription factors. The different temporal window could be explained by the fact that while legume seeds are large and nutrient-rich [63] which means that legume seedlings can afford the carbon expense required for nodule formation, seeds of *D. glomerata* are similar in size to those of *Arabidopsis*, and greenhouse plantlets do not nodulate before they have reached a height of at least 7 cm (K. Pawlowski, unpublished observations).

In a trial experiment using roots of 30-day-old *D. glomerata* seedlings, the exogenous application of 50 nM PAA did not lead to any changes in expression of either *CYCLOPS* or *NIN1* in 24 h treatments (results not shown). To explore the possibility that root development could have a pivotal influence on the transcription of *CYCLOPS* and *NIN1* in assays carried with PAA, a time course experiment was conducted to test the effects of exposure for 2 h, 8 h, and 24 h but, this time, in 54-day-old roots. This assay indicated that the temporal dynamics of *NIN1* expression in response to PAA may be linked to and depend on the age of the plant, meaning that ideally the treatments should be performed on roots of ~54-day-old seedlings over a 24 h period (p = 0.013; S2 Table; S4 Fig). This result was then reproduced in two independent series of experiments showing induction of *NIN1* in PAA-treated roots at concentrations as low as 10 nM (p = 0.008) and 50 nM (p = 0.006) (S2 Table; Fig 5). Consistently, 50 nM PAA exerted a positive effect on *NF-YA1* transcription (p = 0.037) (S2 Table; Fig 5). Although neither BAP nor NAA exerted an effect on the expression of *CYCLOPS*, *NIN1*, or *NF-YA1*, the changes observed on the expression of their respective marker genes *ARR9* (p = 0.036, 10 nM BAP; p = 0.008, 50 nM BAP) and *GH3.1* (p = 0.2, 10 nM NAA; p = 0.002, 50 nM NAA) convincingly reflected the effects of these phytohormones (S2 Table; Fig 5). Collectively, these findings reinforce the idea that, in *D. glomerata*, *GH3.1* represents a marker gene to address responses to NAA, while *SAUR1* represents a marker gene for responses to PAA (p = 0.061, 10 nM PAA; p<0.001, 50 nM PAA; S2 Table). More important, these findings suggest that *DgNIN1* mRNAs were translated, and their products were able to induce the expression of their presumable target gene, *DgNF-YA1*.

## Addition of BAP or inhibition of ethylene or gibberellic acid biosynthesis abolished the induction of *NIN1* by 50 nM PAA

In an attempt to understand the effects caused by the removal of ethylene and gibberellic acid (GA), inhibitors of their biosynthesis were introduced into the experiments, namely L-alpha-(2-aminoethoxy vinyl) glycine (AVG) and paclobutrazol (PBZ), which inhibit, respectively, key enzymes for ethylene and GA biosynthesis. Experiments without added inhibitor (control group) were included to interpret the crosstalk between ethylene or GA signalling and auxin and cytokinin signalling. The cytokinin BAP was used at 10 nM, the natural auxin PAA was used at 50 nM, and, in the combined treatment, 10 nM of BAP were mixed with 50 nM of PAA.

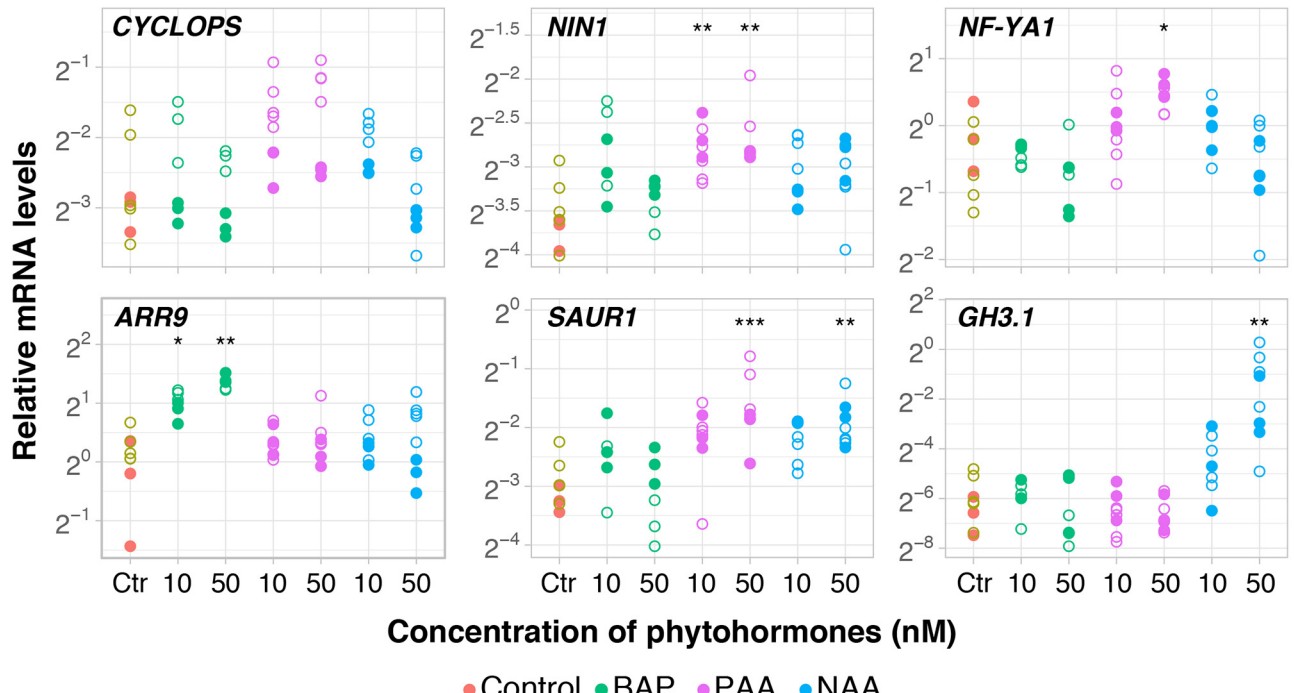

**Fig 5.** *Datisca glomerata* **transcriptional response to cytokinin (BAP) and auxins (PAA and NAA).** The data was collected from two independent experiments (empty and full circles). Transcript abundance was analysed by RT-qPCR after 52-day-old roots treated during 24h (technical replicates, n = 3). *Y*-axis shows the mRNA quantity relative to that of *PUQ* and *TIP41*. Phytohormones and molarities are given on the *X*-axis (number of biological replicates: n = 8 for Ctr, P10, and N50; n = 7 for P50 and N10; n = 6 for B10 and B50). Significant differences to the control are highlighted at p<0.05 (*) and p<0.01 (**). Gene names are given.

The effects exerted by BAP in this experiment were not as prominent as in previous assays, however BAP effects were still moderately at play based on the expression of *ARR9* in the control group (p = 0.057) (S2 Table; Fig 6). *SAUR1* expression was significantly induced by PAA in the control group (p<0.01), but this induction was abolished in the presence of BAP (p = 0.081), AVG (p = 0.18) or PBZ (p = 0.25) (S2 Table; Fig 6). Consistent with these results, expression of *NIN1* (p = 0.015) and *NF-YA1* (p = 0.006) was also induced by PAA in the control group, but not when BAP was added as well (*NIN1*, p = 1.0; *NF-YA1*, p = 1.0), or when ethylene (*NIN1*, p = 0.77; *NF-YA1*, p = 0.089) or GA (*NIN1*, p = 0.31; *NF-YA1*, p = 1.0) were depleted (Fig 6; S2 Table). It is worthwhile to note that roots treated with PBZ for 11 days before analysis showed a significant decrease in mRNA abundance of *CYCLOPS* (p = 0.031) when compared with their control and AVG-treated counterparts (Fig 6).

In summary, ethylene and gibberellin were required for the induction of *SAUR1*, *NIN1* and *NF-YA1* expression by PAA, while BAP abolished the induction of the expression of these three genes by PAA (Fig 7). In combination with the negative effect of BAP on the expression of *CYCLOPS*, *NIN1* and *NF-YA1* in 30-day-old plantlets (Fig 4), these results suggest that in contrast with legumes, cytokinin acts negatively on the nodulation of *D. glomerata*.

## Discussion

With this study we sought to (i) gain qualitative insight in the involvement of phytohormones in the developmental program of actinorhizal nodules of *Datisca glomerata*, (ii) quantify the impact of phytohormones on transcription of early transcription factors in nodule

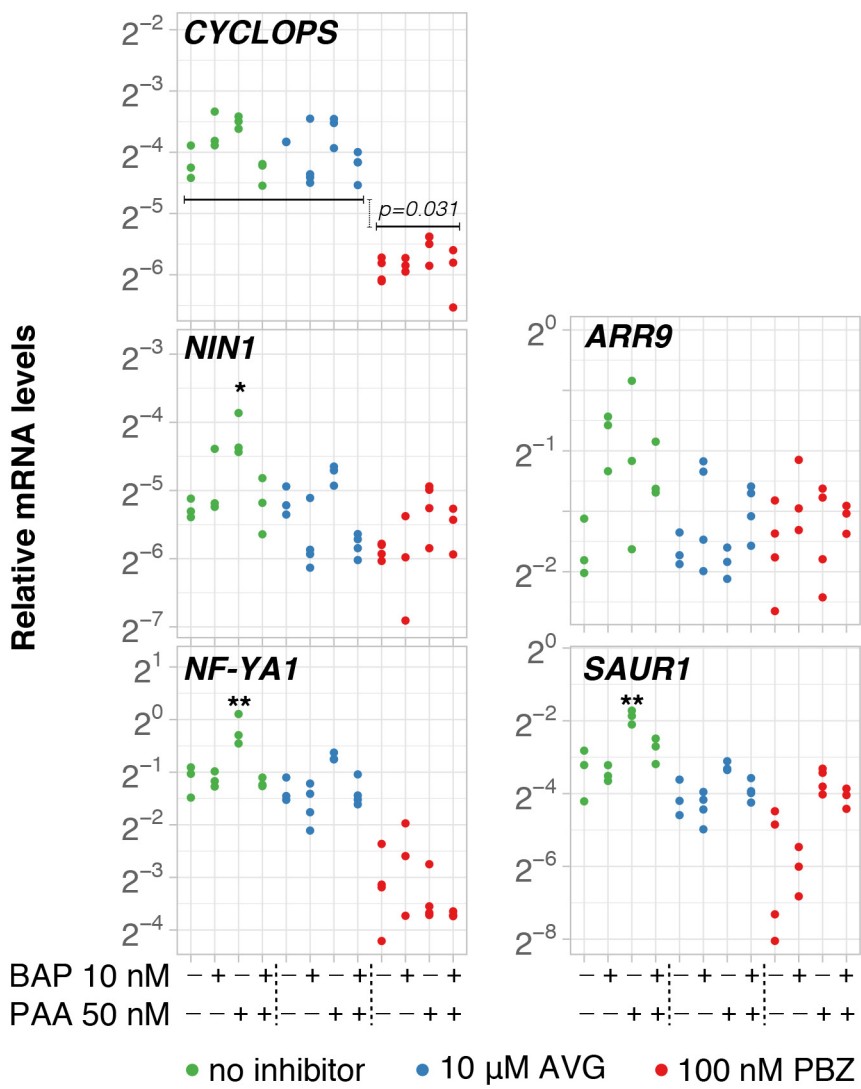

**Fig 6.** ***Datisca glomerata*** **transcriptional response to BAP, PAA, and the co-treatment BAP+PAA under the presence or absence of inhibitors for ethylene (AVG) or gibberellin biosynthesis (PBZ).** Transcript abundance was analysed by RT-qPCR after 52-day-old roots treated during 24h (technical replicates, n = 3). With the exception of BAP10+AVG, BAP10+PAA50+AVG, Ctr+PBZ, and PAA50+PBZ (n = 4), the number of biological replicates was n = 3. *Y*-axis shows the mRNA quantity relative to that of *TIP41*. Phytohormones and molarities are given on the *X*-axis (+, present; -, absent). Significant differences to the control are highlighted at p<0.05 (*) and p<0.01 (**) after *Tukey* HSD test. Gene names are given.

development and (iii) build up knowledge on the prospect that the *D. glomerata* NF-YA1 promoter is, as is the case in model legumes, targeted by the transcription factor *NODULE INCEPTION* (*DgNIN1*; [55]). From here, we set out to understand the regulation of *DgNIN1* and *DgNIN2*; even though no functional studies have been carried out for either *NIN* gene, their relevance for root nodulation has been established by comparative phylogenomics and phylotranscriptomics studies [3, 64]. *In silico* analysis of *PACE* sequences from *DgNIN1* and *DgNIN2* promoters support the possibility of protein-DNA interactions (Fig 2B and 2C), a prediction that is line with the findings of Cathebras et al. [54] that have shown that the *PACE* elements of both *DgNIN* genes suffice for induction by the *Lotus japonicus* CCaMK/CYCLOPS

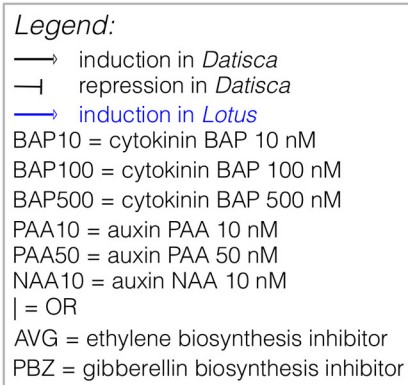
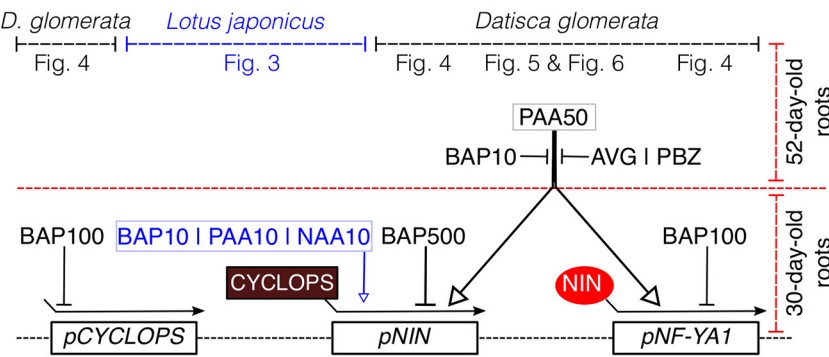

**Fig 7. Summary of the main findings reported in this manuscript.** Positive and negative effects of exogenously added phytohormones auxin (PAA and NAA), cytokinin (BAP), or inhibition of endogenous biosynthesis of ethylene or gibberellins are shown for *Datisca glomerata* (black) and *Lotus japonicus* (blue) on the expression of key genes encoding transcription factors involved in nodule development (*CYCLOPS*, *NIN*, and *NF-YA1*) in 30- and 52-old-day seedlings (see red dashed line). Information about the Figure in the manuscript from which the evidence was drawn is provided on top. CYCLOPS and NIN proteins are respectively highlighted in maroon and red.

complex; indeed, *DgNIN1* combined with its *PACE* element could complement the nin15 mutant of *L. japonicus*, meaning that this non-standard *PACE* element conferred induction by the CCaMK/CYCLOPS complex of *L. japonicus* [54]. Thus far no functional studies were performed on CYCLOPS of actinorhizal plants, however protein 3D models analysis indicate that the distinct features of DgCYCLOPS might correlate with a larger plasticity of the protein [65], in comparison with CYCLOPS of other actinorhizal plants or those of legumes (Fig 2D and 2E). Altogether, there is no reason to exclude the possibility that DgCYCLOPS might play a pivotal role by transactivating both *DgNIN1* and *DgNIN2*, but evidence about these events has yet to be provided.

The induction of *NIN* by exogenously applied cytokinin had been demonstrated for the model legumes *Lotus japonicus* [17] and *Medicago truncatula* [18] using seedlings growing on agar in Petri dishes. As proof of concept for our hydroponic system, we took advantage of the knowledge transpired from these studies to address the response of *NIN* transcription to exogenously applied synthetic cytokinin 6-Benzylaminopurine (BAP) and synthetic auxin 1-Naphthaleneacetic acid (NAA) in roots of *L. japonicus*, using *L. japonicus Ckx3* cytokinin oxidase/dehydrogenase3 as a marker gene for the cytokinin response [61], and *Gretchen Hagen3* (*GH3*; [62]) as marker gene for the auxin response. In line with the findings of Heckmann et al. [17], we showed that *LjNIN* expression could be induced using BAP and NAA in the hydroponic system (Fig 3). Additionally, we showed that the expression of *LjNIN* could also be induced in a time- and dosage-dependent manner by the natural auxin Phenylacetic Acid (PAA), although PAA could not be detected in roots of *L. japonicus* [66]. Interestingly, expression of the auxin response marker gene *GH3.1* was upregulated by NAA, but not by PAA. This divergence suggests that some auxin effects are likely to have been overlooked in *L. japonicus*.

The *LjNIN* promoter is activated by the CYCLOPS/CCaMK complex [37]. Since the expression of *LjCYCLOPS* was neither induced by BAP, NAA nor by PAA in this study, we conclude that the induction of *LjNIN* expression did not require an increase in *CYCLOPS* mRNA. The induction of *LjNIN* by cytokinin had been linked to the well-established fact that cytokinin signaling occurs in the root cortex during legume nodule organogenesis [15]. The induction of *LjNIN* by auxin was not reported previously and is likely to be linked to the accumulation of

auxin at the onset of nodule induction as well as to the fact that NIN recruits the lateral root development program for nodule organogenesis [25, 67–69].

When establishing marker genes to track responses to NAA, PAA, and BAP in *D. glomerata*, the results were in line with those observed previously for *L. japonicus*: the expression of the ortholog of the cytokinin-induced type A response regulator gene *ARR9* from *Arabidopsis* [70] was upregulated by BAP in *D. glomerata*, while the expression of the *D. glomerata* ortholog of *LjGH3*, *DgGH3.1*, was upregulated by NAA. *DgSAUR1*, a member of another group of genes known for auxin responsiveness [71], was identified and served as marker gene for PAA effects. PAA had been discovered as auxin-type phytohormone in 1935 [72–74]; however, because experiments on pea and oat showed that the activity of PAA was much lower than that of IAA, PAA was not examined in detail. The synthesis of PAA does not follow the same pathway as that of IAA [75], and it does not show basipetal transport like IAA [76, 77]. In spite of this, IAA and PAA effects on gene expression showed some overlap when examined for *Arabidopsis*, *e.g.*, four members of the *Arabidopsis* GH3 family were induced by both IAA and PAA (*GH3.2*, *GH3.3*, *GH3.4* and *GH3.5*), while all *Arabidopsis* SAUR genes reacting to auxin were only induced by IAA, not by PAA [77]. It should be pointed out, however, that Sugawara et al. [77] examined gene expression already after one hour of exposure to auxin and used much higher auxins concentrations, namely 1 μM IAA and 10 μM PAA– 20 or 200 times higher, respectively, than the concentration established as optimal for the induction of *LjNIN/DgNIN1* expression in this study, 50 nM for both NAA and PAA. At any rate, these results showed that the differential effects of NAA and PAA on gene expression during nodule development require further examination.

As discussed above, the induction of legume *NIN* expression by exogenously applied cytokinin had been ascribed to the fact that cytokinin signalling takes place in the root cortex in the course of legume nodule organogenesis, leading to the formation of the legume nodule primordium [17, 18]. On the other hand, shoot-derived cytokinin transported in the phloem has been implicated in the systemic repression of nodulation during the autoregulation of nodulation [78]. Legume nodule primordia are induced in the root cortex and the root pericycle, while actinorhizal nodule primordia are induced in the root pericycle [46]. Therefore, we do not necessarily expect the same involvement of phytohormones in both processes. For induction of an organ primordium in the root pericycle, close to the auxin maximum at the protoxylem pole [79], auxin would be expected as inducer. Indeed, the results of this study showed unambiguously that the expression of *D. glomerata NIN1* is induced by the natural auxin PAA, though not by the synthetic auxin NAA, and that the induction by PAA is abolished in the presence of the synthetic cytokinin BAP. The difference between PAA and NAA effects on *DgNIN1* expression is consistent with the facts that (i) PAA is the dominant auxin in *D. glomerata* roots [51] and (ii) NAA is unlikely to replace PAA in every context since both auxins were shown to be transported differently in *Arabidopsis* [77]. Intriguingly, when 50 nM PAA and 10 nM BAP were applied simultaneously, the induction of expression of *DgNIN1* as well as that of *DgSAUR1* by PAA was abolished (Fig 6). This was a striking finding that, for *DgNIN1*, can be discussed from two angles: first, it reinforces the idea that BAP is indeed acting as a negative regulator of *DgNIN1* expression, but treatment with 10 nM BAP alone does not seem to suffice to reduce *DgNIN1* expression in 30-day-old plants. At this age, as shown in Fig 4, BAP has the strongest negative effect on the expression of *DgCYCLOPS*, in concentrations equal to or above 50 nM, while it only affected *DgNIN1* expression at 500 nM. Combined, the data suggest that in the presence of exogenous BAP, two synergistic effects for negative regulation of *DgNIN1* expression occur: one, direct, via downregulation of *DgNIN1* transcription, the other, indirect, abolishing the induction by PAA. In this context, it is interesting to note that the response of *DgCYCLOPS* expression to 50 nM PAA was inconsistent throughout our

experiments, ranging from non-induced (Fig 5, p = 0.19; Fig 6, p = 0.12) to induced (S4 Fig; p = 0.013). This raises the possibility that the discrepancies observed were linked to variable amounts of endogenous auxins or cytokinins (S5A Fig).

The response difference between 30-day-old *D. glomerata* seedlings to phytohormones compared to 52- or 54-day-old seedlings can be explained by the fact that nodulation requires a considerable carbon investment which has to be supplied from seeds or from photosynthesis. Legume seedlings are routinely nodulated in Petri dishes, *i.e.*, very young, given that they obtain most of their nutrients from their large seeds. This is also possible for seedlings of the actinorhizal shrub *Coriaria myrtifolia* which forms seeds similar in size to those of *L. japonicus* [80]. However, in 30-days-old *D. glomerata* seedlings grown from seeds of 0.8–1 mm diameter, nodulation has to be downregulated before they have produced enough leaves to provide photosynthates for additional strong carbon sinks (nodules), a process that would likely abolish or at least weaken the effects of exogenous application of phytohormones on *DgNIN1* expression, as was observed in this study–no effect of PAA was found, and BAP only acted at 500 nM.

Altogether, our results support the observations of Gauthier-Coles et al. [81] for the actinorhizal tree *Alnus glutinosa* who showed that here, cytokinin (*cis*-zeatin, *trans*-zeatin, dihydrozeatin, and kinetin were used) does not act as positive regulator of pseudonodule development as it does in legumes. Rodríguez-Barrueco et al. [82], on the other hand, found pseudonodule development in response to treatment with two different cytokinins, 2-isopentenyl adenin (2IP) and kinetin; however, the latter study was not performed under axenic conditions, which means that other phytohormones produced by bacteria or fungi may have been involved.

Regarding *DgSAUR1* repression, the available evidence suggests that the induction of *DgSAUR1* expression by exogenous PAA is abolished by the addition of exogenous BAP, which might be connected to the presence of a dozen of *cis*-regulatory motifs for cytokinin response in its promoter and which may bind repressor(s) (S5B Fig). Interestingly, the hypothesis that cytokinin is downregulating *DgSAUR1* expression is similar to the situation observed for *L. japonicus GH3* in presence of 100 nM BAP (Fig 3), further emphasizing that the interplay between auxins and cytokinins must be considered when using these promoters as markers.

To test the effects of ethylene, *D. glomerata* plantlets were grown on media supplied with an inhibitor of ethylene biosynthesis, 10 µM of L-alpha-(2-Aminoethoxyvinyl) glycine (AVG). Inhibition of ethylene biosynthesis had a negative effect on *DgNIN1* expression (Fig 6). Similarly, *DgSAUR1* was not induced by exogenous PAA In the presence of AVG (Fig 6). In short, the effect of AVG in combination with PAA resembled the effect of the combination of BAP and PAA. This effect could be ascribed to increased amounts of cytokinins as in legumes, ethylene negatively regulates cytokinin accumulation during nodule induction [19]. However, it has to be mentioned that in the actinorhizal species *Casuarina glauca*, whose nodule primordia form in the root pericycle like in *D. glomerata*, ethylene has a negative effect on nodulation like in legumes [83]. Another possible explanation could involve the role of ethylene in the transport of auxin from the shoot to the nodulation site, shown to result in high auxin accumulation [34] and implying that endogenous auxin levels in AVG-treated roots could be below the level required for the induction of *DgNIN1*. At any rate, the fact that the inhibition of ethylene biosynthesis abolished the induction of *DgNIN1* expression by PAA in *D. glomerata* roots (p = 0.77) was the second relevant difference found between *D. glomerata* and legumes with regard to phytohormone effects on *NIN* expression [33]. It is relevant to point out that studies of the effects of ethylene on legume nodulation were normally not performed in liquid culture, but on seedlings growing on solid agar. While ethylene has been found as a requirement for nodulation of *Sesbania rostrata* under waterlogged conditions [84], it inhibited nodulation under non-flooded conditions in the same species [85]. This suggests that the positive role of ethylene in the regulation of *DgNIN1* expression might be a feature of nodulation under

waterlogged conditions, or for nodulation that does not involve the formation of infection threads in root hairs, as is the case for *D. glomerata* [46]. However, later studies have shown that nodulation via root hairs in *L. japonicus* involves the production of ethylene by the plant [53].

For the inhibition of gibberellin biosynthesis, the growth conditions had to be modified to enable root development before exposure to the inhibitor paclobutrazol (PBZ); PBZ was added after 41 days of seedling growth. This pre-treatment with PBZ was necessary because the inhibition of gibberellins biosynthesis during the 24 h exposure to auxin/cytokinin would not affect existing gibberellins [86]. Thus, the possibility that a growth-retardant effect caused by PBZ interfered with the experiment cannot be ruled out, given the significant differences in the effect on *DgNIN1* expression observed between roots assays on day 30 *vs*. day 52. At any rate, in the presence of PBZ, PAA did not cause significant changes in the expression levels of any gene examined, including that of *SAUR1* (p = 0.25; Fig 6). Yet, in the presence of PBZ, the expression of *DgCYCLOPS* was significantly reduced (Fig 6), suggesting that the induction of *DgCYCLOPS* expression involves a DELLA protein. Combined, these results are consistent with the positive involvement of gibberellin signalling in legume nodule organogenesis mediated by the GRAS transcription factor DELLA1 as shown in *Medicago truncatula* [24].

Overall, these findings indicate that the symbiotic program for nodule induction in *D. glomerata* significantly differs from that of *L. japonicus* (Fig 7). *LjNIN* could be induced directly by application of Nod factors [17] as well as directly by exogenous supply of BAP and NAA, and later also by PAA, in a time- and dosage-dependent manner (Fig 3). The data obtained with the model legume *L. japonicus* confirm the well-known role played by cytokinin signaling towards nodule organogenesis in the root cortex of legumes, a divergent trait that has probably emerged ~60 Mya ago in nodulating species of legumes but is absent in non-nodulating legumes as well as in non-legumes, including actinorhizal plants [81, 87]. In contrast, in *D. glomerata*, the expression of *DgNIN1* was induced by PAA, an effect that was abolished in the presence of cytokinin (Fig 6), arguing for a pivotal role of auxin in the nodulation of *D. glomerata* regarding the regulation of *DgNIN1*, and in turn *DgNF-YA1*. In analogy to legumes, local auxin accumulation–equivalent to cytokinin accumulation in legumes as a result of rhizobial Nod factor signaling–might be induced *via Frankia* signalling, maybe involving ethylene on the plant side (Figs 5 and 6), inducing *DgNIN1* expression and, consequently, nodule organogenesis.

The possibility that the expression of *DgNIN1* is linked to a role in the regulation of lateral root formation, as was found for lipochitooligosaccharide Nod factor-responsive *NIN* genes of legumes and poplar [88], should be considered, however the fact that the expression of *DgNIN1* cannot be induced in plantlets too young to nodulate (Fig 4) and it shows high levels of expression in nodules (Fig 2) make it clear that *DgNIN1* has to play a relevant role in nodulation. Since the standard *cis*-regulatory element *PACE* of *DgNIN2* is involved in infection thread progression [54], a role for *DgNIN1* and its uncommon *PACE* in nodule organogenesis can be hypothesized, however functional characterization studies are required to unravel the role played by DgCYCLOPS in this context. As it stands, it is tempting to speculate that DgCYCLOPS plays a pivotal role in the nodule development program of *Datisca glomerata* by driving nodule organogenesis via *DgNIN1* and *Frankia* accommodation via *DgNIN2*. At this point, however, it is unclear whether the regulation of *CYCLOPS* in roots by phytohormones observed in this study is relevant for nodulation, or only for arbuscular mycorrhization. If the former, it would have to involve the induction of the expression of other transcription factor genes as *DgNIN2* expression was not detected in roots under the conditions described in this study. Altogether, the data obtained in this study suggest that the ancient program for

symbiosis established between *Frankia* and *D. glomerata*, whose nodules originate from the root pericycle, requires auxin to induce the expression of *DgNIN1* and *DgNF-YA1*.

## Conclusions

An axenic hydroponic system was established to examine the effects of exogenously applied phytohormones on the expression of key genes required for nodule organogenesis on roots of the actinorhizal plant *Datisca glomerata* and on those of the model legume *Lotus japonicus*, which we used as a control. Marker genes for phytohormone perception were established: *DgARR9* served as marker for the synthetic cytokinin BAP, whilst *DgGH3.1* and *DgSAUR1* served as markers for the synthetic and natural auxins NAA and PAA, respectively. The collected data showed that *DgNIN2*, whose promoter contains a standard *PACE cis*-regulatory element, is not expressed in roots, whereas *DgNIN1*, whose promoter harbors a non-standard *PACE*, showed induction by the auxin PAA, but not by the cytokinin BAP. In contrast, *LjNIN* expression was induced by BAP, NAA, and, when measured 24 h after application of 10 nM, by PAA. The induction of *DgCYCLOPS*, *DgNIN1* and *DgNF-YA1* transcription by PAA was abolished when either BAP, the ethylene biosynthesis inhibitor AVG or the gibberellin biosynthesis inhibitor PZB were added together with PAA, suggesting that the induction by PAA required certain levels of ethylene and gibberellin, but could be abolished by low levels of exogenously applied cytokinin BAP. Altogether, the phytohormone involvement in the induction of nodules in the actinorhizal species *D. glomerata* differs from that of legumes concerning the roles of auxin, cytokinin and ethylene, while the role of gibberellin has been likely conserved across lineages.

## Material and methods

### Plant material

*Datisca glomerata* (C. Presl) Baill seeds were originally obtained from plants in Gates Canyon, Vacaville, CA, USA [89]. Plants for seed production were cultivated in a greenhouse in a 1:1 (v/v) mixture of germination soil (Weibull Trädgard AB, Hammenhög, Sweden) and sand (1–2 mm Quartz; Rådasand AB, Lidköping, Sweden). Light conditions in the greenhouse were 150–300 $\mu Em^{-1}\,s^{-1}\,m^{-2}$ at 13 h light/22 ˚C and 11 h dark/19 ˚C. Experiments involving the model legume *Lotus japonicus* were performed with ecotype *Gifu*.

### Hydroponic system

To use roots challenged with exogenous hormones in downstream gene expression studies, an axenic hydroponic system was developed (Fig 1). Irrespective of the plant species investigated, seeds were surface-sterilized [5 min in 70% ethanol, 0.1% SDS; 20 min in 2% sodium hypochlorite, 0.05% SDS; rinsed 4 times in sterile MilliQ water] and imbibed in sterile MilliQ water at 4ºC, during 5 days in the dark. After thorough washing, seeds were disposed in 20 mm x 20 mm metal grids and allowed to germinate in a thin layer of ¼ strength Hoagland's solution [90] supplemented with 10 mM nitrogen (Hoagland's $N^+$) and 0.8% plant agar (Duchefa, The Netherlands) (Fig 1A). One week later, metal grids containing germinated seeds were placed over 18 ml vials filled with liquid Hoagland's $N^+$. Note that the employed strategy enables recently emerged root tips to develop within a fully controlled environment. The bioassay proceeded in Magenta boxes (Sigma-Aldrich, Germany) under continuous white light for 53±1 days for *D. glomerata* and 30 days for *L. japonicus*. Light conditions in the growth chamber were 200 $\mu E\,s^{-1}\,m^{-2}$ at 16h light/23˚C and 8h dark/18˚C.

Treatments used in this study included exogenous applications of the phytohormones cytokinin 6-Benzylaminopurine (BAP; Duchefa cat. no. B0904), the natural auxin Phenylacetic Acid (PAA; Sigma-Aldrich cat. no. P6061), and the synthetic auxin 1-Naphthaleneacetic Acid (NAA; Sigma-Aldrich cat. no. N0640) in combination or not with antagonists of ethylene or gibberellic acid (GA) biosynthesis. Equal molarities of the solvent sodium hydroxide (for BAP) and ethanol (for PAA and NAA) were added in control assays.

For inhibition of ethylene biosynthesis, *D. glomerata* plantlets were grown on Hoagland's N$^+$ supplied with 10 µM of the inhibitor L-alpha-(2-aminoethoxy vinyl) glycine (AVG; Sigma-Aldrich cat. no. 32999) for 52 days before being transferred to phytohormone-containing medium or mock control medium. For inhibition of GA biosynthesis, *D. glomerata* plantlets were grown for 41 days on Hoaglands N$^+$. Then, plantlets were transferred to Hoagland's N$^+$ supplied with 100 nM of the GA inhibitor Paclobutrazol (PBZ; Sigma-Aldrich cat. no. 43900) and grown for 10 days before being transferred to phytohormone-containing medium or mock control medium.

All phytohormone treatments were carried out in ¼ strength Hoagland's solution without nitrogen (Hoagland's N$^-$). Time treatments ranged between experiments from 2h to 24h. Upon treatment, roots were shock-frozen in liquid nitrogen and stored at -80˚C. Special attention was paid during this step to avoid mechanical disturbance that could lead to the synthesis of jasmonates as previously reported [91]. Frozen roots were used for isolation of total RNA and subsequent gene expression analysis.

## Bioinformatic analysis: Identification of orthologs, promoter comparative analysis, and protein structure prediction

To identify putative legume orthologs, the transcriptome of *D. glomerata* was searched using as a query previously characterized proteins from model legumes [56]. The survey relied on reciprocal Blast searches combined with HMMER (v. 3.2.1) models [92] and top candidate proteins were then used for phylogenetic reconstruction.

To select sequences for phylogenetic reconstruction of CYCLOPS, the orthologous group ENOG410IGIK (http://eggnogdb.embl.de) was searched at UniProtKB (E-value = 1e-200) and truncated proteins were removed (total = 71). For NF-YA1, the *M. truncatula* protein [93] was queried at UniProtKB and the HMMER profile built after the top 66 hits was used in a second search into the same database (E-value = 1.5e-59), rendering 93 curated proteins. To these protein sets, candidate orthologs from *D. glomerata* and *Ceanothus thyrsiflorus* were added. Sequences were aligned using ProbCons version 1.12 [94]. Aligned positions were selected with BMGE using the BLOSUM62 substitution matrix [95]. Phylogenetic trees were estimated using RAxML v.8.2.10 [96] using the "PROTGAMMAAUTO" model and rapid bootstrapping where bootstrap replicates were automatically stopped upon convergence with autoMRE bootstopping [97].

Evaluation of secondary order effects of *PACE cis*-regulatory elements placed on promoter regions of *DgNIN1* and *DgNIN2* was carried out by deepDNAshape with the number of layers set to 4 and feature set to propeller-twisting of base-pairs (ProT) [98].

Three-dimensional model structures of CYCLOPS were predicted at the AlphaFold database [99, 100].

## RNA isolation and gene expression analysis (RT-qPCR)

Root tips grown hydroponically were mechanically disrupted using a TissueLyser II (Quiagen, Germany) over 3 min at 28 Hz. Roots and nodules of greenhouse-grown *D. glomerata* were macerated in liquid nitrogen. Macerated tissues were immediately used for total RNA isolation

with an on-column DNase treatment (Sigma Aldrich, Spectrum Total RNA isolation kit). The extraction method yielded high quality RNA (RIN>7.8 measured in a 2100 Bioanalyzer; Agilent Genomics). Prior to cDNA synthesis, a second step of gDNA removal was performed on 0.5–1.0 μg of total RNA using the Heat&Run kit from ArticZymes (Tromsø, Norway). Total RNA was reverse transcribed in a final volume of 20 μl following the instructions for the TATAA GrandScript cDNA synthesis kit (TATAA Biocenter, Sweden). cDNA preparations were diluted $10^{-1}$ and 2 μl each were used as templates in 10 μl PCR reactions in the presence of 1x Maxima SYBR green (Thermo Fisher Scientific, Lithuania) supplied with 300 nM of each primer in an Eco Real Time PCR instrument (Illumina, USA). PCR conditions were as described in Zdyb et al. [91]. Controls for gDNA-derived copies and primer dimer assessment were considered by the inclusion of water as a template, RT-minus runs, and examination of melting dissociation curves. For all the experiments, statistical analyses were carried out based on at least three biological replicates with a minimum of two technical PCR repeats. Primers were designed using Primer3 at NCBI Primer-Blast server and are listed in S1 Table.

## Gene expression data: Considerations for normalization and statistics

To identify stable reference genes in roots of *D. glomerata*, a subset of the genes listed by Czechowski et al. [101] was used for a systematic validation of the RT-qPCR data generated in this study. In a pilot assay, the stability of six reference candidate genes was compared using Norm-Finder [102]. The pilot assay rendered three stable genes for *L. japonicus* and for *D. glomerata*. Genes considered as a reference for roots of *D. glomerata* were: *Ubiquitin carboxyl-terminal hydrolase 5*, *TIP41*, and *Elongation factor 1-α* (GenBank accessions are provided in S1 Table). Reference genes for roots of *L. japonicus* were *Ubiquitin-conjugating enzyme* [91], *F-type H⁺-transporting ATPase subunit beta* [60], and *Protein Phosphatase 2A* [60]. For both *D. glomerata* and *L. japonicus*, NormFinder stabilization indices of these three reference genes were calculated in every assay to identify the most stable internal normalizer (or combination of normalizers). Demonstrative exponential phase Cq values were then considered to calculate the normalization factors of target genes by $2^{-\Delta\Delta Cq}$ [103]. Differences in gene expression were assessed by analysis of variance (ANOVA). Equality of variances was evaluated by Levene's test. For equal variances, one-way ANOVA was fitted to log$_2$-transformed $2^{-\Delta\Delta Cq}$ with treatments as fixed term and biological repeats as random effects accounting for dependencies between treatments; to estimate and adjust for pairwise multiple comparisons, a non-parametric post-hoc Tukey HSD test was applied. For unequal variances, a Welch's pairwise t-test with Holm's correction was used. For the assay involving the interaction of two factors (treatments *vs*. AVG treatment), a two-way ANOVA was fitted to the data with "treatments" as a fixed factor in the model. Biological repeats accounted for random effects nested under the interaction term. Statistics were performed using R Statistical Software (v4.2.3; [104]) and are summarized in S2 Table.

## Supporting information

**S1 Fig. Identification of auxin reporter genes for *Datisca glomerata*.** (A) A member of the Small <u>A</u>uxin-<u>U</u>p <u>R</u>NA family of *D. glomerata* (DgSAUR1) shows high sequence similarity with members of the orthologous group ENOG410J0BD along with proteins from *Parasponia andersonii* (PON39119.1) and *Medicago truncatula* (KEH36239.1). (B) Members of the *Gretchen Hagen 3* family are expressed in nodules of *D. glomerata*; DgGH3.1 and DgGH3.2 were compared by multiple sequence alignment with twelve sequences from the orthologous group ENOG410IHQ2 (B). To this set, previously characterized sequences from *Arabidopsis thaliana* were added: AtGH3.5/WES1 and AtGH3.1/JAR1. While AtGH3.5/WES1 has high

affinity for both indole-3-acetic and salicylic acid (Westfall et al., 2016), AtGH3.11/JAR1 responds to jasmonic acid (Westfall et al., 2012). Illustrated blocks represent individual motifs to which a link of structural arrangement with function has been demonstrated (refer to Figures 2 and 3 of Westfall et al., 2016). Additional metrics like conservation, quality, consensus, and occupancy are given. (C) Genetic dependencies of these promoters to auxins on 54-day-old *D. glomerata* roots. *Y-axis* shows the mRNA quantity relative to that of *PUQ*, *EF1-α*, and *TIP41*. Phytohormones and molarities are given on the *X-axis*. Significant differences to the control are highlighted by Welch's pairwise t-test with Holm's correction at $p<0.01$ (**). Gene names are given.
(PDF)

**S2 Fig.** Full alignment of CYCLOPS from 8 nodulating species and 1 non-nodulator (*Trema tomentosa*) (A). For *Datisca glomerata*, regions spanning disorder predicted (purple) and coil (light green) domains are depicted by a horizontal top bar. Note the presence of a disorder predicted domain at the N-terminus of *Datisca glomerata* that is only conserved in *Ceanothus thyrsiflorus* and *Alnus glutinosa*. Maximum-likelihood phylogenetic reconstruction of CYCLOPS/IPD3 (B). Multiple protein sequence alignment was produced after searching the orthologous group ENOG410IGIK at UniProtKB. Model legume orthologs are highlighted in pink, *Ceanothus thyrsiflorus* in green (GenBank accession MN388817), and *Datisca glomerata* in blue.
(PDF)

**S3 Fig. Rooted tree showing the maximum-likelihood phylogenetic reconstruction of NUCLEAR FACTOR YA1 (NF-YA1).** Nomenclature is from UniProtKb. Proteins from model legumes were highlighted in pink, *Ceanothus thyrsiflorus* in green (GenBank accession MN388814), and *Datisca glomerata* in blue.
(PDF)

**S4 Fig. Temporal dynamics of *CYCLOPS* and *NIN*1 expression in roots of 54-day-old *Datisca glomerata* plantlets.** Transcript abundance was analysed by RT-qPCR after 2h, 8h, and 24h treatments with 10 and 50 nM of PAA (n = 3 for both technical and biological replicates). Abundance of target mRNA is given relative to that of *PUQ* and *EF1-α*. *X*-axis shows nanomolarities of PAA. *Y*-axis depicts fold changes compared to control roots. Welch's pairwise t-test with Holm's correction highlight differences at $p<0.05$ (*) and $p<0.01$ (**). Gene names are given.
(PDF)

**S5 Fig. Potential *cis*-regulatory elements found 2000 bases upstream of (a) *DgCYCLOPS* and (b) *DgSAUR1* coding regions, known to be involved in auxin and cytokinin responses.**
(PDF)

**S1 Table. Genes analysed in this study; primers used.**
(XLSX)

**S2 Table. Results of Welch t-test to compare expression levels between roots and nodules of *Datisca glomerata*.**
(XLSX)

## Acknowledgments

The authors are very grateful to Max Griesmann and Martin Parniske (Department of Genetics, Ludwig Maximilians University of Munich, Germany) for multiple helpful discussions

about genes and phytohormones involved in the nodulation of legumes. The support from Science for Life Laboratory, the National Genomics Infrastructure, NGI, and UPPMAX (UPPNEX project ID b2013247) computational infrastructure is gratefully acknowledged.

## Author Contributions

**Conceptualization:** Marco Guedes Salgado, Katharina Pawlowski.

**Formal analysis:** Marco Guedes Salgado, Daniel Lundin.

**Funding acquisition:** Katharina Pawlowski.

**Investigation:** Marco Guedes Salgado, Pooja Jha Maity.

**Methodology:** Marco Guedes Salgado.

**Project administration:** Katharina Pawlowski.

**Supervision:** Daniel Lundin, Katharina Pawlowski.

**Writing – original draft:** Marco Guedes Salgado.

**Writing – review & editing:** Marco Guedes Salgado, Pooja Jha Maity, Daniel Lundin, Katharina Pawlowski.

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
