## [Decision Letter · Decision Letter 0]

10 Jun 2024

PONE-D-24-19931Phenylacetic acid induces NIN expression in hydroponically treated roots of the actinorhizal plant Datisca glomerataPLOS ONE

Dear Dr. Pawlowski,

Thank you for submitting your manuscript to PLOS ONE. After careful consideration, we feel that it has merit but does not fully meet PLOS ONE’s publication criteria as it currently stands. Therefore, we invite you to submit a revised version of the manuscript that addresses the points raised during the review process.

We look forward to receiving your revised manuscript.

Kind regards,

Rajappa Janyanaik Joga, PhD

Academic Editor

PLOS ONE

Journal Requirements:

“Swedish Research Council Vetenskapsrådet VR2012-03061 to KP

Carl Tryggers Stiftelse för Vetenskaplig Forskning CTS13:354 to KP”

Reviewers' comments:

Reviewer's Responses to Questions

**Comments to the Author**

1. Is the manuscript technically sound, and do the data support the conclusions?

Reviewer #1: Partly

Reviewer #2: No

Reviewer #3: Yes

Reviewer #4: Yes

2. Has the statistical analysis been performed appropriately and rigorously? 

Reviewer #1: Yes

Reviewer #2: No

Reviewer #3: N/A

Reviewer #4: Yes

3. Have the authors made all data underlying the findings in their manuscript fully available?

Reviewer #1: Yes

Reviewer #2: Yes

Reviewer #3: Yes

Reviewer #4: Yes

4. Is the manuscript presented in an intelligible fashion and written in standard English?

Reviewer #1: Yes

Reviewer #2: Yes

Reviewer #3: Yes

Reviewer #4: Yes

5. Review Comments to the Author

Reviewer #1: In comparison with legumes, genes invovled in actinorhizal nodule organogenesis remain less understood. In this study, the authors analyzed the responses of several nodulation genes in Datisca glomeratac hydroponic system.This study supports the view that cytokinin signaling is central for cortex-induced nodules of L. japonicus but acts as an antagonist to the induction of pericycle-induced nodules of D. glomerata by PAA The results would be potentially interesting to the field and teh findings are interesting. I have several specific comments:

1. The authors showed candidate gene expression in response to phytohormone treatment . However, no data was presented how the phytohormones affect nodule number or formation. Physiological observation should be performed regarding the nodulation phenotype in different treatments.

2. The picture of the hydroponic culture should be moved from supplemental figure to Fig.1.

3. Phylogenetic trees should be shown for the genes used in this study, demonstrating that these are indeed orthologs of nodulation genes in legumes.

4.The agricultural significance of legume and actinorhizal nodulation should be introduced in the first paragraph. For example, nodulation in legumes reduce fertilizer usage and improves soybean yield (https://doi.org/10.1038/s41477-024-01696-x ).

Reviewer #2: In the manuscript Salgado et al., describes the primary hormonal response in primary symbiotic specific transcription factors after treatment with phytohormone. The major take home massage of this paper is that in the actinorhizal plant Datisca glomerata the gene expression of NIN induces after auxin (PAA) treatment instead of CK (BAP) treatment. This is an important massage for the nodule biologist. Despite this fact, there are some important lacunae in the manuscript that need to be addressed.

Major points:

NIN1 expresses in root and eventually induces in nodule where as NIN2 is nodule specific. Further, In the figure 1 where both NIN1 and NIN2 expression has been shown there it is clear that NIN2 expression is higher than NIN1. If so, then why the complete study is conducted only using NIN1. Only the proximal promoter analysis can not be used as an logic to not conduct the qRT-PCR analysis of NIN2.

Minor points.

1. In the page 3 in introduction the involvement of the phytohormone para written in an confusing manor.

2. Page 4, ‘suggesting that DgNIN1 signaling might dependent on auxin or cytokinin’. There should be auxin and/or cytokinin.

3. Page5. Preliminary assays confirmed ARR9 as a suitable marker gene for cytokinin in D. glomerata, owing to an increase of ARR9 mRNA levels in the presence of BAP (discussed below), please add figure number here.

4. In Fig 1, the statistics is not marked as mentioned in the legend. For all the figure please mention the number of technical and biological replicates.

5. The transcriptional profile exhibited by CYCLOPS in this assay indicates that its transcription is tightly regulated by BAP levels within the range of 10 nM to 100 nM. Why the CRE1 cytokinin receptor gene was not taken into study? I suggest to check CRE1 expression as well.

6. The expression of L. japonicus cytokinin oxidase/dehydrogenase3 (LjCkx3; [56]) was induced by BAP when applied for a period of 8h. In the Fig 2, if auxin and BAP are antagonists then why CKX3 expression did not decrease in PAA, NAA treatment? Please elaborate

7. D. glomerata roots exhibits a different temporal window to react to possible effects of cytokinin when compared with roots of L. japonicus, which could be best explained by differences in root development between the two species. Please elaborate few lines about root developmental difference between these two species.

8. PBZ-treated roots showed a significant decrease in mRNA abundance of CYCLOPS and NF-YA1 when compared with their control and AVG-treated counterparts. The significance was not marked in the Fig 5.

Discussion

9. Since the induction of the actinorhizal nodule primordium takes place in the root pericycle [42], the same mechanism would not necessarily be expected for actinorhizal plants. Please elaborate this and connect with the provided evidences from this study.

10. Indeed, the results of this study showed unambiguously that the expression of D. glomerata NIN1 is repressed by the synthetic cytokinin BAP and induced by the natural auxin PAA, though not by the synthetic auxin NAA. How the LjNIN and DgNIN1 is receiving this application of exogenous PAA and NAA in different style? Can you add some comments about its signalling?

Reviewer #3: Comments

The study conducted by Salgado et al, focused on the genes and phytohormones requirements during the nodulation program of the actinorhizal plant Datisca glomerata, a largely unknown process. The purpose of the study is clear and very well written. This working model represents numerous challenges, that limit the approaches employed in model legumes. Nonetheless, they obtained interesting results on the differential responses of D. glomerata roots on symbiotic gene expression, after treatment with synthetic phytohormones, specially cytokinin. Undoubtedly this work provides valuable information to the community interested in plant-microbes mutualistic associations.

Minor comments

Strictly talking, the title of the manuscript is correct, but probably not very “catchy”. The authors could consider a title that includes the terms hormones and remove Phenylacetic acid. The different phytohormone-related gene expression might be highlighted.

Page 1. “ In all these interactions, the microsymbionts are not vertically transmitted”. In common bean and probably other legumes, rhizobia can be vertically transmitted (https://doi.org/10.1111/j.1574-6941.1998.tb00513.x)

Results: The length of the subsections can be shortened

Figure Legends: Please include the number of harvested plants per biological replicate.

Cytokinin and cytokinin-related genes seem to be differentially required in “unusual” nodulation programs, for instance the nodulation in Parasponia (doi: 10.3389/fpls.2018.00284) and the intercellular infection program in Lotus japonicus (https://doi.org/10.1093/plphys/kiaa049). The authors could discuss more about it.

Reviewer #4: The present study focuses on hormonal regulation of genes encoding key regulators of symbiotic nodules development in actinorhizal plant Datisca glomerata. Molecular mechanisms underlying the symbiotic nodule development and its hormonal control is well established for legume-rhizobia symbiosis; however, for actinorhizal plants, its regulation remains less understood. This study is aimed to fill this gap. Overall, the results of this study are interesting and propose that key regulators of symbiosis are regulated differently in legumes and actinorhizal plant D. glomerata: notably, the expression of the DgNIN gene is induced by auxin (phenylacetic acid), but not by cytokinin BAP as it was shown for L. japonicus and other model legumes. However, the obtained results require further clarification and discussion.

Comments:

1. The inhibitory effect of BAP on expression levels of CYCLOPS, NIN1 and NF-YA1 was shown for 30-day-old plants. However, at this age, in a trial experiment using roots of 30-day-old D. glomerata seedlings, the exogenous application of 50 nM PAA did not lead to any changes in expression of either CYCLOPS or NIN1 in 24h treatments. Therefore, authors suggested the possibility that root development could have a pivotal influence on the transcription of CYCLOPS and NIN1 at this stage (30-day-old D. glomerata seedlings). In 54-day-old seedlings, 10 and 50 nM PAA induced NIN1 expression, however, in this experiment, 10 and 50 nM BAP did not downregulate NIN1 activity at this stage (10 nM BAP even tends (although non-significantly) to increase NIN1 expression, Figure 4). Therefore, the conclusion made by the authors (throughout the Discussion section) that BAP inhibits NIN expression in D. glomerata is not quite correct. These findings should be discussed more accurately. The effect of BAP and auxin treatment on gene expression levels in D. glomerata should be compared at the same developmental stage.

2. Expression of NIN1 is induced in nodules compared to roots, while NIN2 is expressed nodule-specifically. Since NIN1 expression is not nodule-specific, can this gene have functions in root development that are not related to the development of symbiotic nodules, but some other functions, possibly related to the development of lateral root? It would be nice to discuss this issue. Why only NIN1 expression was checked, not NIN2, which is nodule-specific?

3. Previously, it was reported that CYCLOPS binding site and CE-region (containing many putative cytokinin response regulator binding sites) are conserved in legume NIN promoters (Liu et al., 2019; Liu and Bisseling. 2020, doi: 10.3390/genes11070777). CYCLOPS/IPD3 binding site is located about 3 kb upstream of the start codon, whereas CE-region is located far upstream from the NIN start codon (in Medicago it is about 18 kb upstream, and in Lotus it is about 45 kb upstream of the NIN start codon) (Liu et al., 2019, DOI: 10.1105/tpc.18.00478 ). It was suggested that the gain of the CYCLOPS binding site and CE-region was essential step to establish NIN expression during nodule formation.

Are the CYCLOPS-binding site and CE-region conserved in NIN1 and NIN2 genes in D. glomerata as well? It should be checked and discussed in the manuscript in order to understand the evolutionary conservatism of NIN regulation in actinorhizal plants.

4. Gauthier-Coles et al., 2019 (DOI: 10.3389/fpls.2018.01901) reported that nodulating legumes are distinguished by a sensitivity to cytokinin in the root cortex leading to pseudonodule development. However, in the study by Gauthier-Coles et al, a nodulating actinorhizal species Alnus glutinosa formed no pseudonodules after application of cytokinin. In contrast, Rodriguez-Barrueco et al., 1973 (DOI: 10.1111/j.1399-3054.1973.tb03107.x) showed that cytokinin (2IP) application induced pseudonodules on A. glutinosa. Therefore, whether exogeneous cytokinin triggers cortical cells in actinorhizal species remains unclear. It would be nice to clarify this and discuss these findings in accordance with the results obtained in the present study.

5. The beginning of Abstract should be re-written in order to make it clearer and more consistent.

6. In conclusion, it is desirable to add a scheme illustrating the hormonal regulation of the expression levels of key nodule regulators in legumes such as L. japonicus and actinorhizal plant D. glomerata in order to summarize the findings of this study.

6. PLOS authors have the option to publish the peer review history of their article (what does this mean?). If published, this will include your full peer review and any attached files.

Reviewer #1: No

Reviewer #2: **Yes: **Senjuti Sinharoy

Reviewer #3: **Yes: **JESUS MONTIEL

Reviewer #4: No

---

## [Author Response · Author response to Decision Letter 0]

13 Aug 2024

Responses to reviewers’ comments

Authors:

First, we would like to thank all reviewers for their efforts. You will see that we had to rewrite the manuscript to some extent – originally, these data were obtained in 2017-2019, and we did not check the promoters of the two NIN genes of D. glomerata in the beginning of the project. The fact that the DgNIN1 promoter contains a non-standard PACE element – which nevertheless works as PACE element, see Cathebras et al. (2022) – has prompted us to include some bioinformatic analysis on both PACE elements from D. glomerata and DgCYCLOPS (added to Fig 2 and S2 Fig).

Reviewer #1: In comparison with legumes, genes invovled in actinorhizal nodule organogenesis remain less understood. In this study, the authors analyzed the responses of several nodulation genes in Datisca glomerata hydroponic system.This study supports the view that cytokinin signaling is central for cortex-induced nodules of L. japonicus but acts as an antagonist to the induction of pericycle-induced nodules of D. glomerata by PAA The results would be potentially interesting to the field and teh findings are interesting. I have several specific comments:

Reviewer #1-1: 

The authors showed candidate gene expression in response to phytohormone treatment . However, no data was presented how the phytohormones affect nodule number or formation. Physiological observation should be performed regarding the nodulation phenotype in different treatments.

Response to #1-1:

This is problematic for the major reasons pointed below:

- Because the microsymbiont of D. glomerata, Candidatus Frankia datiscae Dg1, cannot be cultured, (a) we have no way to quantify the inoculum, and (b) we have to infect with crushed nodules of older plants, which means we have a non-sterile inoculum. We have to work in the greenhouse.

- Datisca seeds are similar in size to Arabidopsis seeds, not as large as the nutrient-rich seeds of legumes. Seedlings do not form nodules. Plants have to be several weeks old (at least 7 cm high) when they are predisposed to be nodulated.

- Nodulation of actinorhizal plants is not as fast as for legumes. We first find nodules ca. 6 weeks after infection. This timing is for Sweden in summer (nodulation in California is faster). Nodulation in Sweden in Winter is slower (8-10 weeks).

Thus, we are dealing with plants in 12 cm diameter pots (Datisca root systems are large). How are we to add the phytohormones – every three days, once per week? What role does the soil microbiome play? How does the phytohormone distribute in the pot? We tried this once before [for Demina et al. (2019) with water vs. NAA; NAA is at least known to be stable] and the standard deviations were enormous; we ended up not including the results in the manuscript. 

Reviewer #1-2:

The picture of the hydroponic culture should be moved from supplemental figure to Fig.1.

Response to #1-2:

Fig S1 has been moved to Fig 1, and the other Figures have been renumbered.

3. Phylogenetic trees should be shown for the genes used in this study, demonstrating that these are indeed orthologs of nodulation genes in legumes.

Response to #1-3:

These trees are already included in the supplement (Supplementary Figures S2-S4 in the previous version, Supplementary Figures S1-S3 in the revised version).

4.The agricultural significance of legume and actinorhizal nodulation should be introduced in the first paragraph. For example, nodulation in legumes reduce fertilizer usage and improves soybean yield (https://doi.org/10.1038/s41477-024-01696-x ).

Response to #1-4:

This has been done: “While legume symbioses are essential in agriculture insofar as they yield protein-rich seeds while rendering their host plants with an independent source of nitrogen fertilizer, thereby contributing to increases of nitrogen pools in surrounding ecosystems, actinorhizal species mostly represent pioneer plants and are often used in reforestation or soil reclamation [Andrews et al. 2011].”.

Reviewer #2: In the manuscript Salgado et al., describes the primary hormonal response in primary symbiotic specific transcription factors after treatment with phytohormone. The major take home massage of this paper is that in the actinorhizal plant Datisca glomerata the gene expression of NIN induces after auxin (PAA) treatment instead of CK (BAP) treatment. This is an important massage for the nodule biologist. Despite this fact, there are some important lacunae in the manuscript that need to be addressed.

Major points:

Reviewer #2-1:

NIN1 expresses in root and eventually induces in nodule where as NIN2 is nodule specific. Further, In the figure 1 where both NIN1 and NIN2 expression has been shown there it is clear that NIN2 expression is higher than NIN1. If so, then why the complete study is conducted only using NIN1. Only the proximal promoter analysis can not be used as an logic to not conduct the qRT-PCR analysis of NIN2.

Response to #2-1:

As pointed out, NIN2 is expressed in a nodule-specific manner, while the experiments for this manuscript were performed on roots. We could not induce nodules in an axenic system since the microsymbiont of D. glomerata, either Candidatus Frankia datiscae or Candidatus Frankia californiensis, is not cultivable. We could not detect NIN2 levels of expression on either roots of greenhouse-grown plants or on roots from our hydroponic system, independent of the presence of exogenous phytohormones. We cannot induce nodules in the hydroponic system, because we do not have axenic inoculum (see Response to #1-1). (Nodulation is possible under hydroponic conditions provided the cultures are aerated, but this requires large cultures started with 6-weeks-old plant, and there is significant growth of non-Frankia bacteria).

Thus, while the promoter of NIN2 contains the PACE element (Cathebras et al. 2022) and thus is induced by the CCaMK/CYCLOPS complex in response to bacterial signalling, it was impossible to examine its induction using plantlets growing in an axenic hydroponic system. 

Cathebras et al. (2022) have shown that the DgNIN2 promoter contains a standard PACE element while the DgNIN1 promoter harbors a non-conventional PACE element (see Fig 2B,C). Nevertheless, experiments of Cathebras et al. (2022) using D. glomerata PACE elements driving the expression of NIN were able to complement and rescue the Lotus japonicus nin15 phenotype. We have added a large paragraph in the Result section of the manuscript about cis-regulatory PACE elements in D. glomerata as well as about their presumable target, DgCYCLOPS. This section was included to get a better understanding on whether PACE-CYCLOPS events in D. glomerata are supported from an in silico perspective. It reads now:

“D. glomerata contains two copies of NIN [3] and although both NIN1 and NIN2 are induced in nodules compared to roots, only NIN2 is expressed nodule-specifically with no detectable levels of expression in roots of either seedlings or greenhouse-grown plants (Fig 2; for NIN1 see also [55]). Cathebras et al [54] have shown the presence of the cis-regulatory element PACE within the promoter of DgNIN2 (termed DgNIN1 in [54]) as a requirement for induction by the CCaMK/CYCLOPS complex, whilst the promoter of DgNIN1 contains an uncommon and extended version of PACE, which compared to that of DgNIN2, or those of other root nodule-forming plants, contains an insertion of eight nucleotides at the core of PACE and displays high nucleotide conservation across PACE-Y and PACE-X, and ca. 67% conservation at the core of the PACE element (Fig 2B). To gain a better understanding of the PACE-CYCLOPS interactions in D. glomerata, we compared the amino acid sequence of DgCYCLOPS with those of CYCLOPS proteins from a range of root nodule-forming plants and one non-nodulator, Trema tomentosa. Alignment of CYCLOPS orthologs showed that the primary structure of CYCLOPS, along with its ortholog in Ceanothus thyrsiflorus, displays a ~40 amino acid sequence extension at its N-terminus that is neither present in Alnus glutinosa, Parasponia andersonni, nor in the legumes included in the analysis (S2A Fig); importantly, the first 20 amino acids represent a disordered domain whose presence at the N-terminus seems to be a feature of actinorhizal plants as it was only found in Datisca glomerata (position 1-20), Alnus glutinosa (position: 1-33) and Ceanothus thyrsiflorus (position 16¬-36) (S2A Fig). Furthermore, 17 amino acids upstream of the largest and well-conserved disordered domain, placed at the C-terminus, the primary structure of DgCYCLOPS shows high dissimilarity with that of other orthologs (Fig 2D; full alignment in S2A Fig). To investigate whether this dissimilarity could lead to alterations in protein secondary structure, AlphaFold structure predictions were carried out and the comparison of the predicted models showed the presence of an α-helix in Datisca glomerata CYCLOPS that is absent in CYCLOPS proteins of Ceanothus thyrsiflorus, L. japonicus, and Medicago truncatula (Fig 2E).”

Minor points.

Reviewer #2-2:

1. In the page 3 in introduction the involvement of the phytohormone para written in an confusing manor.

Response to #2-2:

We have rewritten the entire part that deals with phytohormones.

Reviewer #2-3:

2. Page 4, ‘suggesting that DgNIN1 signaling might dependent on auxin or cytokinin’. There should be auxin and/or cytokinin.

Response to #2-3:

We apologize for the mistake. This has been fixed now.

Reviewer #2-4:

3. Page5. Preliminary assays confirmed ARR9 as a suitable marker gene for cytokinin in D. glomerata, owing to an increase of ARR9 mRNA levels in the presence of BAP (discussed below), please add figure number here.

Response to #2-4:

This has been fixed: “Preliminary assays confirmed ARR9 as a suitable marker gene for cytokinin in D. glomerata, owing to an increase of ARR9 mRNA levels in the presence of BAP (see below in Fig. 4).”

Reviewer #2-5:

4. In Fig 1, the statistics is not marked as mentioned in the legend. For all the figure please mention the number of technical and biological replicates.

Response to #2-5:

We apologize for the omission. This has been fixed now.

Reviewer #2-6:

5. The transcriptional profile exhibited by CYCLOPS in this assay indicates that its transcription is tightly regulated by BAP levels within the range of 10 nM to 100 nM. Why the CRE1 cytokinin receptor gene was not taken into study? I suggest to check CRE1 expression as well.

Response to #2-6:

Our interest was focussed on the common symbiotic signalling pathway that leads to Calcium spiking, activation of CCaMK, activation of CYCLOPS and finally, activation of NIN. We included one of the next genes in the cascade, NF-YA1, to confirm NIN transcription. We discussed the inclusion of other genes when we had the first PAA results. CRE1 was eliminated early in the game because (a) we not know whether CRE1 is involved in the nodulation program of actinorhizal plants in general and Datisca glomerata in particular (see van Zeijl et al. – it is not even clear whether it is involved in nodulation in Parasponia), and we had/have neither the time nor the (wo)manpower to perform CRISPR/Cas9 hairy root experiments, (b) the induction of CRE1 transcription by exogenously applied cytokinin in Medicago truncatula is rather weak. Furthermore, our results in this study do not point at cytokinin accumulation as a factor in nodule organogenesis in D. glomerata. At any rate, the main objective was to find a reliable molecular readout for BAP, and ARR9 fitted that need pretty well as it behaved in a rather consistent manner throughout our experiments.

Reviewer #2-7:

6. The expression of L. japonicus cytokinin oxidase/dehydrogenase3 (LjCkx3; [56]) was induced by BAP when applied for a period of 8h. In the Fig 2, if auxin and BAP are antagonists then why CKX3 expression did not decrease in PAA, NAA treatment? Please elaborate.

Response to #2-7:

Fig 3 (Fig 2 in the previous version) does not show the effects of combinations of phytohormones. According to our results, BAP acts as antagonist of PAA in D. glomerata. It should not act as an antagonist of NAA or PAA in L. japonicus since here, all three phytohormones induce NIN expression when applied individually.

Reviewer #2-8:

7. D. glomerata roots exhibits a different temporal window to react to possible effects of cytokinin when compared with roots of L. japonicus, which could be best explained by differences in root development between the two species. Please elaborate few lines about root developmental difference between these two species.

Response to #2-8:

We have inserted this text:

“The different temporal window could be explained by the fact that while legume seeds are large and nutrient-rich [Mathesius 2022] which means that legume seedlings can afford the carbon expense required for nodule formation, seeds of D. glomerata are similar in size to those of Arabidopsis, and greenhouse plantlets do not nodulate before they have reached a height of at least 5 cm (K. Pawlowski, unpublished observations).”

Reviewer #2-9:

8. PBZ-treated roots showed a significant decrease in mRNA abundance of CYCLOPS and NF-YA1 when compared with their control and AVG-treated counterparts. The significance was not marked in the Fig 5.

Response to #2-9:

The reduction of expression levels as a consequence of PBZ treatment is significant for CYCLOPS, but not for NF-YA1. This has been corrected in Fig 6 (Fig 5 in the previous version).

Discussion

Reviewer #2-10

9. Since the induction of the actinorhizal nodule primordium takes place in the root pericycle [42], the same mechanism would not necessarily be expected for actinorhizal plants. Please elaborate this and connect with the provided evidences from this study.

Response to #2-10:

The sentence has been replaced with: “Legume nodule primordia are induced in the root cortex and the root pericycle, while actinorhizal nodule primordia are induced in the root pericycle. Therefore, we do not necessarily expect the same involvement of phytohormones in both processes.”

Reviewer #2-11:

10. Indeed, the results of this study showed unambiguously that the expression of D. glomerata NIN1 is repressed by the synthetic cytokinin BAP and induced by the natural auxin PAA, though not by the synthetic auxin NAA. How the LjNIN and DgNIN1 is receiving this application of exogenous PAA and NAA in different style? Can you add some comments about its signalling?

Response to #2-11:

We would like to stress that the antagonism between BAP and PAA works not only for the induction of DgNIN1, but also for the induction of DgSAUR1. However, since NAA and PAA were shown to be transported differently in Arabidopsis (Sugawara et al. 2015), we do not expect that a PAA antagonist will necessarily work as an NAA antagonist. PAA is the dominant auxin in D. glomerata (Demina et al. 2019); its distribution in other actinorhizal plants has yet to be examined. PAA could not be detected in Lotus japonicus at all (Ng and Mathesius 2022), so we feel that for the examination of PAA effects, L. japonicus is not the ideal model system.

Reviewer #3: Comments

The study conducted by Salgado et al, focused on the genes and phytohormones requirements during the nodulation program of the actinorhizal plant Datisca glomerata, a largely unknown process. The purpose of the study is clear and very well written. This working model represents numerous challenges, that limit the approaches employed in model legumes. Nonetheless, they obtained interesting results on the differential responses of D. glomerata roots on symbiotic gene expression, after treatment with synthetic phytohormones, specially cytokinin. Undoubtedly this work provides valuable information to the community interested in plant-microbes mutualistic associations.

Response: We thank the reviewer for these positive remarks!

Minor comments

Reviewer #3-1:

Strictly talking, the title of the manuscript is correct, but probably not very “catchy”. The authors could consider a title that includes the terms hormones and remove Phenylacetic acid. The different phytohormone-related gene expression might be highlighted.

Response to #3-1:

We would rather keep the reference to PAA 

---

## [Decision Letter · Decision Letter 1]

18 Oct 2024

PONE-D-24-19931R1The auxin phenylacetic acid induces NIN expression in the actinorhizal plant Datisca glomerata, whereas cytokinin acts antagonisticallyPLOS ONE

Dear Dr. Pawlowski,

Thank you for submitting your manuscript to PLOS ONE. After careful consideration, we feel that it has merit but does not fully meet PLOS ONE’s publication criteria as it currently stands. Therefore, we invite you to submit a revised version of the manuscript that addresses the points raised during the review process.

We look forward to receiving your revised manuscript.

Kind regards,

Rajappa Janyanaik Joga, PhD

Academic Editor

PLOS ONE

Reviewers' comments:

Reviewer's Responses to Questions

**Comments to the Author**

1. If the authors have adequately addressed your comments raised in a previous round of review and you feel that this manuscript is now acceptable for publication, you may indicate that here to bypass the “Comments to the Author” section, enter your conflict of interest statement in the “Confidential to Editor” section, and submit your "Accept" recommendation.

Reviewer #5: All comments have been addressed

Reviewer #6: All comments have been addressed

2. Is the manuscript technically sound, and do the data support the conclusions?

Reviewer #5: Yes

Reviewer #6: Yes

3. Has the statistical analysis been performed appropriately and rigorously? 

Reviewer #5: Yes

Reviewer #6: Yes

4. Have the authors made all data underlying the findings in their manuscript fully available?

Reviewer #5: Yes

Reviewer #6: Yes

5. Is the manuscript presented in an intelligible fashion and written in standard English?

Reviewer #5: Yes

Reviewer #6: Yes

6. Review Comments to the Author

Reviewer #5: Manuscript PONE-D-24-19931R1

After carefully reading the revised version, I came to know that the authors have successfully incorporated all the suggested changes and fully addressed the comments/changes proposed by honorable reviewers. I do not have much comments but only few suggestions that can be considered for further improvement.

It is really embarrassing to review the manuscript having no line number.

I have only few questions

1- How cytokinin and auxin interact in regulating DgNIN1 expression in Datisca glomerata?

2- What is the role of ethylene and gibberellic acid in PAA-induced expression of DgNIN1?

3- How do nodule formation processes in Datisca glomerata differ from those in Lotus japonicus?

Conclusion

Usually, the conclusion is the key part of any manuscript, the authors has provided the refence, there is no need of reference. Conclusion of the present must be supported by strong statements justifying and summarizing the results and need to add future prospective.

References:

Some references are too old, I suggest updating.

Figures: I would strongly suggest replacing the figures with original ones having high resolution.

Reviewer #6: In the present work, the author focuses on the regulation of genes encoding

key regulators of symbiotic nodules development in the actinorhizal plant Datisca

glomerata by the phytohormones phenylacetic acid (PAA) or Benzylaminopurine (BAP). The mechanisms underlying the symbiotic nodule development and its hormonal control are well established for legume-rhizobia symbiosis at the molecular level; however, for actinorhizal plants, its regulation remains poorly understood. In my opinion, the results of this study are interesting and propose that key regulators of symbiosis are differentially regulated in legumes and actinorhizal, in particular, the expression of the DgNIN gene, which is induced by PAA, but not by BAP as it was shown for the model legumes Lotus japonicus. The manuscript was previously reviewed, and in my opinion, the authors addressed the concerns raised by the reviewers very well; now, the results are presented and discussed adequately.

However, minor concerns must be addressed:

- I suggest using “possess” instead of “go back to” in the abstract and introduction.

- The term "soil recovery" is more appropriate than "soil reclamation" in the introduction.

- The foot of Figure 2 is not correctly indicated: A) It refers to the level of transcripts; B) Refers to the PACE Cis-regulatory elements in the promoter of NIN1 and NIN2…

- Reference to the figure 2A is missing in the results section.

- Figure S1 is the same as figure S4. Correct the references to S4 Fig in the results section.

- The S4 Fig footnote does not correspond to S4 Fig. S4 Fig is missing.

- The S5 Fig footnote is not necessary.

7. PLOS authors have the option to publish the peer review history of their article (what does this mean?). If published, this will include your full peer review and any attached files.

Reviewer #5: No

Reviewer #6: **Yes: **Homero Reyes de la Cruz

---

## [Author Response · Author response to Decision Letter 1]

8 Nov 2024

Responses to reviewers’ comments Manuscript PONE-D-24-19931R1

Reviewer #5 comment 1:

After carefully reading the revised version, I came to know that the authors have successfully incorporated all the suggested changes and fully addressed the comments/changes proposed by honorable reviewers. I do not have much comments but only few suggestions that can be considered for further improvement.

It is really embarrassing to review the manuscript having no line number.

I have only few questions

1- How cytokinin and auxin interact in regulating DgNIN1 expression in Datisca glomerata?

Response to #5-1:

We do not know how the signal transduction pathways of auxin (here, PAA) and cytokinin interact during nodule induction on D. glomerata. We can only discuss our results in the context of previous studies carried out on legume mutants and the information on the nodule physiology of D. glomerata available to date. NIN was, based on results obtained on legumes (Soyano et al. 2015, doi: 10.1093/pcp/pcu168; Shen and Feng 2024 doi: 10.3389/fpls.2023.1284720) and the actinorhizal species Casuarina glauca (Clavijo et al. 2015 doi: 10.1111/nph.13506), recruited for infection and nodule organogenesis. Since in contrast with legume nodule primordia, nodule primordia of D. glomerata are induced in the pericycle, placed at the vicinity of protoxylem poles and where auxin levels are expected to be higher, it is not surprising that auxin, and in particular the most abundant auxin in D. glomerata, PAA, plays a pivotal role regarding NIN expression which should be required for nodule organogenesis.

We had written in the discussion:

“…the induction of legume NIN expression by exogenously applied cytokinin had been ascribed to the fact that cytokinin signalling takes place in the root cortex in the course of legume nodule organogenesis, leading to the formation of the legume nodule primordium [17,18]. On the other hand, shoot-derived cytokinin transported in the phloem has been implicated in the systemic repression of nodulation during the autoregulation of nodulation [78]. Legume nodule primordia are induced in the root cortex and the root pericycle, while actinorhizal nodule primordia are induced in the root pericycle [46]. Therefore, we do not necessarily expect the same involvement of phytohormones in both processes.”

We have added now:

“For induction of an organ primordium in the root pericycle, close to the auxin maximum at the protoxylem pole [de Smet et al. 2007], auxin would be expected as inducer.”

Reviewer #5 comment 2:

2- What is the role of ethylene and gibberellic acid in PAA-induced expression of DgNIN1?

Response to #5-2:

The roles of ethylene and gibberellic acid are well examined in legumes (exogenously supplied ethylene inhibits the common symbiotic signalling pathway, Oldroyd et al. 2001, doi: 10.1105/TPC.010193); for actinorhizal plants, however, we can only refer to some data on the effect of exogenous application of ethephon on the nodulation of Casuarina glauca (Casuarinaceae, Fagales; Ngom et al. 2020 doi: 10.1007/978-1-0716-0142-6_9) which does not belong to the same order as D. glomerata (Cucurbitales) and has a different infection mechanism for Frankia (namely via root hairs). Ethephon (which at a pH of 5 and above decomposes to form ethylene), when applied exogenously at a concentration of 50 µM, inhibits the nodulation of C. glauca. Thus, the negative effect of exogenously supplied ethylene on NIN expression in legumes as well as C. glauca is not present in D. glomerata. 

We wrote in the discussion (previous version):

“In short, the effect of AVG in combination with PAA resembled the effect of the combination of BAP and PAA. This effect could be ascribed to increased amounts of cytokinins as in legumes, ethylene negatively regulates cytokinin accumulation during nodule induction [19]. Another possible explanation could involve the role of ethylene in the transport of auxin from the shoot to the nodulation site, shown to result in high auxin accumulation [34] and implying that endogenous auxin levels in AVG-treated roots could be below the level required for the induction of DgNIN1. At any rate, the fact that the inhibition of ethylene biosynthesis abolished the induction of DgNIN1 expression by PAA in D. glomerata roots (p=0.77) was the second relevant difference found between D. glomerata and legumes with regard to phytohormone effects on NIN expression [33].”

We have now added the information on C. glauca:

“In short, the effect of AVG in combination with PAA resembled the effect of the combination of BAP and PAA. This effect could be ascribed to increased amounts of cytokinins as in legumes, ethylene negatively regulates cytokinin accumulation during nodule induction [19]. However, it has to be mentioned that in the actinorhizal species Casuarina glauca, whose nodule primordia form in the root pericycle like in D. glomerata, ethylene has a negative effect on nodulation like in legumes [Ngom et al. 2020]. Another possible explanation could involve the role of ethylene in the transport of auxin from the shoot to the nodulation site, shown to result in high auxin accumulation [34] and implying that endogenous auxin levels in AVG-treated roots could be below the level required for the induction of DgNIN1. At any rate, the fact that the inhibition of ethylene biosynthesis abolished the induction of DgNIN1 expression by PAA in D. glomerata roots (p=0.77) was the second relevant difference found between D. glomerata and legumes with regard to phytohormone effects on NIN expression [33].”

No data on gibberellic acid and actinorhizal nodules are available; in legumes gibberellins inhibit infection thread formation but promote nodule organogenesis (McAdam et al. 2018, doi: 10.1093/jxb/ery046). A DELLA protein, a transcription factor that represses the reaction to gibberellin, is part of the CYCLOPS/CCaMK complex (Pimprikar et al. 2016, doi: 10.1016/j.cub.2016.01.069), a conserved module in arbuscular mycorrhization and legume nodulation, and thus expected to be conserved in actinorhizal nodulation as well, as actinorhizal plants use the common symbiotic signalling pathway (shown for C. glauca and D. glomerata, Gherbi et al. 2008, doi: 10.1073/pnas.0710618105; Markmann et al. 2008, doi: 10.1371/journal.pbio.0060068) and contain NIN genes (Griesmann et al. 2018, doi: 10.1126/science.aat1743) the function of which was proven for C. glauca (Clavijo et al. 2015, doi: 10.1111/nph.13506). When gibberellin binds its receptor, the receptor binds DELLA and causes its degradation, thereby activating gibberellin responses. In Lotus japonicus, gibberellin induces NIN expression directly via a cis-acting element in the NIN promoter (Akamatsu et al. 2021, doi: 10.1111/tpj.15128). Thus, we expected an influence of gibberellin on actinorhizal nodulation due to DELLA being part of the conserved module connecting the common symbiotic signalling pathway and NIN expression, but we could not predict the direction of the influence as the organogenesis of legume and actinorhizal nodules differs. It turned out that in D. glomerata, gibberellin was required for the induction of NIN expression by PAA.

We wrote in the discussion about the effects of the inhibitor of gibberellin biosynthesis, PBZ:

“At any rate, in the presence of PBZ, PAA did not cause significant changes in the expression levels of any gene examined, including that of SAUR1 (p=0.25; Fig 6). Yet, in the presence of PBZ, the expression of DgCYCLOPS was significantly reduced (Fig 6), suggesting that the induction of DgCYCLOPS expression involves a DELLA protein. Combined, these results are consistent with the positive involvement of gibberellin signalling in legume nodule organogenesis mediated by the GRAS transcription factor DELLA1 as shown in Medicago truncatula [24].”

(Since a direct effect of gibberellins on NIN expression as shown for the LjNIN promoter could not be tested in our system, we did not discuss/cite Akamatsu et al. 2021.)

Reviewer #5 comment 3:

3- How do nodule formation processes in Datisca glomerata differ from those in Lotus japonicus?

Response to #5-3:

Nodule formation in D. glomerata has not been studied in detail since the microsymbiont is not cultivable; without a quantifiable inoculum, time course analysis is difficult. At any rate, D. glomerata nodule primordia are induced in the root pericycle and lead to the formation of nodule lobes with a central vascular system, while legume nodule primordia are induced in the root cortex and, in case of L. japonicus, lead to the formation of a determinate nodule without persistent meristem. Since our study focuses on the induction of NIN expression and does not touch on nodule organogenesis, we mention only the differences in the location of the nodule primordium in D. glomerata vs. L. japonicus roots.

Reviewer #5 comment 4:

Conclusion

Usually, the conclusion is the key part of any manuscript, the authors has provided the refence, there is no need of reference. Conclusion of the present must be supported by strong statements justifying and summarizing the results and need to add future prospective.

Response to #5-4:

We fully rewrote the conclusion, which now reads as:

“Conclusion

An axenic hydroponic system was established to examine the effects of exogenously applied phytohormones on the expression of key genes required for nodule organogenesis on roots of the actinorhizal plant Datisca glomerata and on those of the model legume Lotus japonicus, which we used as a control. Marker genes for phytohormones were established: DgARR9 served as marker for the synthetic cytokinin BAP, whilst DgGH3.1 and DgSAUR1 served as markers for the synthetic and natural auxins NAA and PAA, respectively. The collected data showed that DgNIN2, whose promoter contains a standard PACE cis-regulatory element, is not expressed in roots, whereas DgNIN1, whose promoter harbors a non-standard PACE, showed induction by the auxin PAA, but not by the cytokinin BAP. In contrast, LjNIN expression was induced by BAP, NAA, and, when measured 24 h after application of 10 nM, by PAA. The induction of DgCYCLOPS, DgNIN1 and DgNF-YA1 transcription by PAA was abolished when either BAP, the ethylene biosynthesis inhibitor AVG or the gibberellin biosynthesis inhibitor PZB were added together with PAA, suggesting that the induction by PAA required certain levels of ethylene and gibberellin, but could be abolished by low levels of exogenously applied cytokinin BAP. Altogether, the phytohormone involvement in the induction of nodules in the actinorhizal species D. glomerata differs from that of legumes concerning the roles of auxin, cytokinin and ethylene, while the role of gibberellin has been likely conserved across lineages.”

Reviewer #5 comment 5:

References:

Some references are too old, I suggest updating.

Response to #5-5:

With regard to original research papers, we have done our best to cite the oldest one that contains the information we have to refer to. If this is about reviews, we could try to find younger ones – we would however like to point out that our review on actinorhizal plants from 2012 was chosen because it covers D. glomerata nodule structure in some detail, and we are not aware of a younger review that does that. We will gladly make an effort to exchange references but could you please point out which ones appear too old to you?

Reviewer #5 comment 6:

Figures: I would strongly suggest replacing the figures with original ones having high resolution.

Response to #5-6:

The figures we submitted are in .tif format and have high resolution. The problem must be due to the process of assembling the manuscript in the editorial manager. 

Reviewer #6: 

In the present work, the author focuses on the regulation of genes encoding key regulators of symbiotic nodules development in the actinorhizal plant Datisca glomerata by the phytohormones phenylacetic acid (PAA) or Benzylaminopurine (BAP). The mechanisms underlying the symbiotic nodule development and its hormonal control are well established for legume-rhizobia symbiosis at the molecular level; however, for actinorhizal plants, its regulation remains poorly understood. In my opinion, the results of this study are interesting and propose that key regulators of symbiosis are differentially regulated in legumes and actinorhizal, in particular, the expression of the DgNIN gene, which is induced by PAA, but not by BAP as it was shown for the model legumes Lotus japonicus. The manuscript was previously reviewed, and in my opinion, the authors addressed the concerns raised by the reviewers very well; now, the results are presented and discussed adequately.

However, minor concerns must be addressed:

Reviewer #6 comment 1:

- I suggest using “possess” instead of “go back to” in the abstract and introduction.

Response to #6-1:

This was changed in both the Abstract and the Introduction of the current version.

Reviewer #6 comment 2:

- The term "soil recovery" is more appropriate than "soil reclamation" in the introduction.

Response to #6-2:

This was changed in the Introduction of the current version.

Reviewer #6 comment 3:

- The foot of Figure 2 is not correctly indicated: A) It refers to the level of transcripts; B) Refers to the PACE Cis-regulatory elements in the promoter of NIN1 and NIN2…

- Reference to the figure 2A is missing in the results section.

Response to #6-3:

A reference to figure 2A is now included:

“…and although both NIN1 and NIN2 are induced in nodules compared to roots, only NIN2 is expressed nodule-specifically with no detectable levels of expression on roots of either seedlings or greenhouse-grown plants (Fig 2A; for NIN1 see also [55]).”

The figure caption was modified as follows:

“Fig 2 | The PACE elements of DgNIN1 and DgNIN2 put in context. (A) Expression profile of genes encoding orthologs of nuclear transcription factors associated with nodule development in Datisca glomerata; transcript abundance was analysed by RT-qPCR in roots (R) and nodules (N) of greenhouse-grown plants and is given relative to that of the housekeeping gene EF1-α (n=3 for both technical and biological replicates); differences between R and N are shown at p<0.001 (student’s t test). (B) In silico analysis of the cis-regulatory element PACE in DgNIN1 and DgNIN2 promoters; representative logo profile of PACE depicted as position weight matrix out of 14 FaFaCuRo species and 2 non-nodulators (Prunus persica and Ziziphus jujuba) along with a comparison of NIN PACE sequences in D. glomerata showing an 8-nucleotide insertion at the core of DgNIN1 PACE. (C) Evaluation of secondary order effects by propeller-twist (ProT) of base-pairs across the PACE landscape of D. glomerata and model legumes. (D) Implications of primary sequence dissimilarity in secondary structure of CYCLOPS shown as 3D models. Partial alignment of CYCLOPS from 8 nodulating species and 1 non-nodulator (Trema tomentosa) shows dissimilarities in a region encompassing 24 residues (see dashed box) (full alignment in S2A Fig). (E) AlphaFold 3D models showing the implications in secondary structure of DgCYCLOPS at the site of the region covered by the dashed box, highlighting the presence of an α-helix in DgCYCLOPS, from 402Q to 412D, which is absent in Ceanothus thyrsiflorus and model legumes.”

Reviewer #6 comment 4:

- Figure S1 is the same as figure S4. Correct the references to S4 Fig in the results section.

- The S4 Fig footnote does not correspond to S4 Fig. S4 Fig is missing.

Response to #6-4:

We apologize for the error. We have now added the true S4 Fig and deleted the duplicate of S1 Fig.

Reviewer #6 comment 5:

- The S5 Fig footnote is not necessary.

Response to #6-5:

Something must have gone wrong during the previous submission; we apologize. We have included the complete S5 Fig now.

---

## [Decision Letter · Decision Letter 2]

2 Dec 2024

The auxin phenylacetic acid induces NIN expression in the actinorhizal plant Datisca glomerata, whereas cytokinin acts antagonistically

PONE-D-24-19931R2

Dear Dr. Pawlowski

We’re pleased to inform you that your manuscript has been judged scientifically suitable for publication and will be formally accepted for publication once it meets all outstanding technical requirements.

Kind regards,

Anwar Hussain

Academic Editor

PLOS ONE

Additional Editor Comments (optional):

Reviewers' comments:

Reviewer's Responses to Questions

**Comments to the Author**

1. If the authors have adequately addressed your comments raised in a previous round of review and you feel that this manuscript is now acceptable for publication, you may indicate that here to bypass the “Comments to the Author” section, enter your conflict of interest statement in the “Confidential to Editor” section, and submit your "Accept" recommendation.

Reviewer #5: All comments have been addressed

Reviewer #6: All comments have been addressed

2. Is the manuscript technically sound, and do the data support the conclusions?

Reviewer #5: Yes

Reviewer #6: Yes

3. Has the statistical analysis been performed appropriately and rigorously? 

Reviewer #5: Yes

Reviewer #6: Yes

4. Have the authors made all data underlying the findings in their manuscript fully available?

Reviewer #5: Yes

Reviewer #6: Yes

5. Is the manuscript presented in an intelligible fashion and written in standard English?

Reviewer #5: Yes

Reviewer #6: Yes

6. Review Comments to the Author

Reviewer #5: Thank you for sharing the revised version.

The authors have successfully incorporated the changes (where suggested) and now I do not have any further concerns and questions, in my opinion, the manuscript can be accepted for warrant publications.

Reviewer #6: In the new version of the manuscript, the authors addressed all the comments raised in the previous submission...

7. PLOS authors have the option to publish the peer review history of their article (what does this mean?). If published, this will include your full peer review and any attached files.

Reviewer #5: No

Reviewer #6: **Yes: **Homero Reyes de la Cruz

---

## [Editor Report · Acceptance letter]

13 Dec 2024

PONE-D-24-19931R2 

PLOS ONE

Dear Dr. Pawlowski, 

I'm pleased to inform you that your manuscript has been deemed suitable for publication in PLOS ONE. Congratulations! Your manuscript is now being handed over to our production team.

Kind regards, 

on behalf of

Dr. Anwar Hussain 

Academic Editor

PLOS ONE